# Multivalent bicyclic peptides are an effective antiviral modality that can potently inhibit SARS-CoV-2

Katherine U. Gaynor [1,6], Marina Vaysburd[2,6], Maximilian A. J. Harman [1], Anna Albecka[2], Phillip Jeffrey[1], Paul Beswick[1], Guido Papa [2], Liuhong Chen [1], Donna Mallery [2], Brian McGuinness[1], Katerine Van Rietschoten[1], Steven Stanway[1], Paul Brear [3], Aleksei Lulla[3], Katarzyna Ciazynska [2], Veronica T. Chang [2], Jo Sharp [4], Megan Neary[4], Helen Box[4], Jo Herriott[4], Edyta Kijak[4], Lee Tatham[4], Eleanor G. Bentley[4], Parul Sharma[4], Adam Kirby[4], Ximeng Han[4], James P. Stewart [4], Andrew Owen [4], John A. G. Briggs [2,5], Marko Hyvönen [3], Michael J. Skynner [1] ✉ & Leo C. James [2] ✉

COVID-19 has stimulated the rapid development of new antibody and small molecule therapeutics to inhibit SARS-CoV-2 infection. Here we describe a third antiviral modality that combines the drug-like advantages of both. *Bicycles* are entropically constrained peptides stabilized by a central chemical scaffold into a bi-cyclic structure. Rapid screening of diverse bacteriophage libraries against SARS-CoV-2 Spike yielded unique *Bicycle* binders across the entire protein. Exploiting *Bicycles'* inherent chemical combinability, we converted early micromolar hits into nanomolar viral inhibitors through simple multimerization. We also show how combining *Bicycles* against different epitopes into a single biparatopic agent allows Spike from diverse variants of concern (VoC) to be targeted (Alpha, Beta, Delta and Omicron). Finally, we demonstrate in both male hACE2-transgenic mice and Syrian golden hamsters that both multimerized and biparatopic *Bicycles* reduce viraemia and prevent host inflammation. These results introduce *Bicycles* as a potential antiviral modality to tackle new and rapidly evolving viruses.

Modern antiviral therapeutics can be broadly divided into two groups – small molecules and biologics. COVID-19 has seen the deployment of both modalities, with varying success in different use cases. Investigated use cases in COVID-19 have included pre- and post-exposure prophylaxis, treatment of mild to moderate disease, severe disease, or critical disease. Antibodies were amongst the first therapeutics to be administered during the pandemic. Convalescent serum, while initially thought to generate useful protection[1], was subsequently shown to

have limited efficacy[2], even in mild to moderate cases[3,4]. Monoclonal antibody cocktails, such as those from Regeneron (casirivimab and imdevimab),Eli Lilly (bamlanivimab and etesevimab), GSK (sotrovimab) and Astrazeneca (tixagevimab and cilgavimab) proved to be effective in trials conducted prior to the emergence of Omicron[5–8] and were granted Emergence Use Authorizations by the FDA for either treatment or pre-exposure prophylaxis depending on the design of the pivotal trials. However, these antibodies work by blocking binding

[1]Bicycle Therapeutics, Portway Building, Granta Park, Cambridge CB21 6GS, United Kingdom. [2]MRC Laboratory of Molecular Biology, Francis Crick Avenue, Cambridge CB2 0QH, United Kingdom. [3]Department of Biochemistry, University of Cambridge, Cambridge CB2 1GA, United Kingdom. [4]University of Liverpool, Crown Street, Liverpool L69 7ZD, United Kingdom. [5]Max Planck Institute of Biochemistry, Martinsried 82152, Germany. [6]These authors contributed equally: Katherine U Gaynor, Marina Vaysburd. ✉e-mail: michael.skynner@bicycletx.com; lcj@mrc-lmb.cam.ac.uk

between the human ACE2 cellular receptor and the receptor binding domain (RBD) of Spike (S), which is one of the most rapidly changing virus:host interfaces. Predictably, emerging variants of concern (VoC) such as Omicron and its sub-lineages have evaded many of these therapeutic approaches resulting in dramatically reduced virus neutralization[9,10]. Selected drug resistance has also been described for monoclonal antibodies like sotrovimab in clinical trials and during use for clinical care[6,11–16]. The problem of viral escape plus high clinical doses requiring expensive and complex production facilities complicate widespread implementation of antibodies.

Small molecule antivirals offer potential advantages over antibody therapeutics. They are cheaper, more stable, and frequently orally bioavailable. During the early months of the pandemic, huge multinational effort was initiated to repurpose already approved antiviral or immunosuppressive drugs. One such drug was remdesivir, a nucleoside analog chain terminator that blocks replication of viral RNA[17]. Initial studies reported clinical improvement after treatment of hospitalized patients with remdesivir[18–21], but larger clinical trials, such as the WHO Solidarity Trial, revealed no reduction in severe COVID-19 symptoms or mortality[21,22]. Remdesivir has been shown to be effective in mild disease[23], however, treatment requires intravenous administration in a hospital setting and current WHO recommendations are for use in severe but not critical disease[24]. Another repurposed antiviral that has been approved for SARS-CoV-2 is molnupiravir (Merck), which is orally bioavailable and also targets RdRp but leads to lethal mutagenesis during RNA replication. Molnupiravir showed promising results in Phase I and II human clinical trials[25,26], but final Phase III findings revealed only modest reductions in hospitalization from 9.7% to 6.8%[27]. A second oral therapeutic, also repurposed, is the viral 3 C protease/M[pro] inhibitor nirmatrelvir (Pfizer), which is combined with ritonavir to boost plasma concentrations via inhibition of cytochrome P450 (CYP) 3A4 mediated metabolism. Interim results for Phase II-III studies showed an 89% reduced hospitalization risk when initiated within three days of symptom onset. Nirmatrelvir uses a covalent nitrile warhead, meaning it is possible for off-target reactions with nucleophilic residues in host proteins[28] or drug-drug interactions, the latter of which prohibit its use in children or those taking other medicines. Overall, the development of small molecules has been considerably slower than antibodies and entirely based on existing chemical templates that were already known and well-characterized as viral inhibitors before the pandemic. As for monoclonal antibodies, multiple routes of resistance escape have been described to emerge under a selective pressure in vitro for small-molecule drugs, and the resistance liability is yet to be fully understood[29].

The therapeutic chronology of SARS-CoV-2 treatment has therefore been an early adoption of biologics, that are effective but expensive and difficult to administer, followed by a slower roll-out of small molecules that are easier to administer but whose efficacy have yet to be demonstrated in randomized clinical trials. An ideal therapeutic would be one that combines the advantages of both modalities whilst minimizing their respective disadvantages. Bicyclic peptides are a novel therapeutic modality that offers that profile. Bicyclic peptides are entropically constrained short peptides stabilized into a bi-cyclic structure using a central chemical scaffold[30]. The bicyclic peptides described here are structurally and chemically unique, comprising cyclized linear peptides with a symmetric, central small-molecule scaffold that forms covalent thioether bonds with three cysteines encoded in the linear peptide sequence[31–35]. We, therefore, use the term *Bicycles* throughout this work to refer to this specific type of bicyclic peptide. When synthesized as small molecules, *Bicycles* offer attractive drug-like properties such as stability in plasma, good pharmacokinetics (compared to other nonbiologics), and ease of manufacture through chemical synthesis[31,36]. *Bicycles* also offer some of the advantages of antibodies, notably the ability to rapidly select binders from large highly diverse genetically-encoded libraries using

bacteriophage display. Antibodies also have long half-lives and although it may not be possible to endow *Bicycles* with comparable longevity, serum persistence can be improved through simple chemical modifications or multimerization[33,34,37,38]. *Bicycles* also have their own unique advantages; they can be easily conjugated together to create multimeric *Bicycles* that have increased affinity through avidity or that access non-competitive inhibitory mechanisms through simultaneous binding to different proteins or epitopes[33,34]. As with any modality, *Bicycles* will have limitations. Some like low oral bioavailability may be addressable through delivery via alternative routes, such as subcutaneous, inhalation, or intra-nasal dosing. Nevertheless, *Bicycles* represent a distinct modality from small molecules and antibodies with a unique combination of useful properties.

Here we show that the combination of advantages intrinsic to *Bicycles* make it an ideal antiviral therapeutic modality. Our findings reveal that the rapid speed with which they can be identified and optimized, via the *Bicycle* screening platform, is compatible with rapid development to address current and emergent viral pathogens. We have exemplified the concept of *Bicycles* as antivirals by generating potent SARS-CoV-2 antiviral molecules de novo, without any prior structural information. Our results describe the identification, in vitro testing, and demonstration of preclinical virological and anti-inflammatory efficacy of SARS-CoV-2 antiviral *Bicycles*.

## Results

### Bicyclic peptide binders can be rapidly selected against SARS-CoV-2 Spike protein

To obtain novel binders against Spike (S), chemically biotinylated S1, S1-RBD, S2, or S trimer recombinant proteins from the Wuhan-Hu-1 strain of SARS-CoV-2 were used as target material in solution-based panning selections using the *Bicycle* bacteriophage (phage) libraries. The libraries consist of linear peptides (10 to 20 amino acids in length) containing 3 cysteines which are cyclized in situ to form thioether-bonded bicyclic peptide libraries[30] (Fig. 1a). After 4 rounds of selection, enriched phage clones were isolated and sequenced. Sequences were grouped by consensus motifs to form families (Table 1 and Supplementary Table 1). The resulting synthesized *Bicycles* were epitope mapped to determine which region of the spike they bound (Fig. 1b). Where there were a number of closely related sequences belonging to the same epitope group, a single canonical sequence (e.g., E2 or E5) was identified as the exemplar for subsequent experiments (Table 1). Epitope mapping at the level of protein domain was performed using surface plasmon resonance (SPR) with immobilized proteins corresponding to S trimer, S1, S2, NTD, or RBD, with ACE2 as a negative control (Fig. 1c and Supplementary Table 2). Hits were characterized by affinity determination against each protein construct. Non-binders were classified as showing no residual binding at the highest concentration tested. To determine whether peptides bound to distinct or overlapping epitopes, we designed a competition AlphaScreen assay. Here, biotinylated *Bicycles* representative of key sequence families were tethered to Streptavidin donor beads in order to interact with Spike-coated Nickel-chelate acceptor beads (Fig. 1d). Untagged *Bicycles* from different families, but that bound the same Spike protein fragments, were added to elucidate which families were competitive with each other (Fig. 1d and Supplementary Table 3). The combination of SPR domain mapping and AlphaScreen competition assays allowed *Bicycle* hits against S to be classified into 12 distinct "epitope bins," of which only two (E8 & E10) could not be unambiguously mapped (Fig. 1b–d). Taken together, this data shows that distinct *Bicycle* binders can be easily and rapidly obtained that target all parts of Spike.

The identification of discrete groups of binders across the S glycoprotein highlights that *Bicycles* are not restricted to a single immunodominant epitope (Fig. 1b). To investigate this further, we modified the AlphaScreen assay to test whether representative anti-S hits from each epitope group competed with hACE2 for RBD binding. Spike-

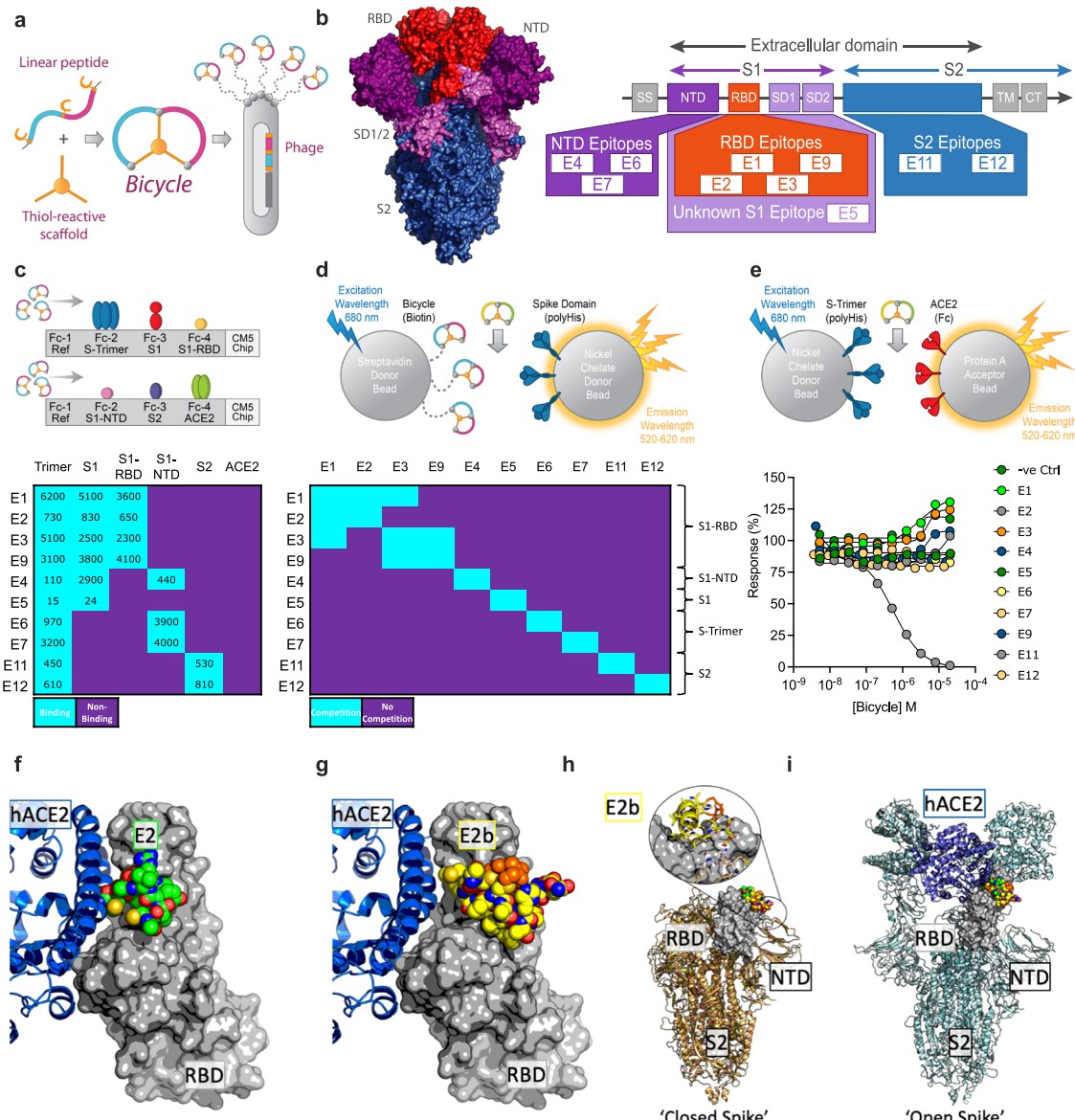

**Fig. 1 | *Bicycles* can target every part of Spike. a** Schematic of *Bicycles* and phage display. *Bicycles* comprise diverse linear peptides split into two variable length polypeptides between three cysteines that when mixed with a small molecule forms a covalent conjugate containing two constrained peptide loops. *Bicycles* were displayed on phage and screened against immobilized Spike (**b**). S1 = S1 subunit, S2 = S2 subunit, SS = Signal sequence, NTD = N-terminal domain, RBD = Receptor binding domain, SD1 = Subdomain 1, SD2 = Subdomain 2, TM = Transmembrane domain, CT = Cytoplasmic tail. *Bicycles* were classified into epitope groups (E1 to E12) based on where they bind Spike. **c** *Bicycles* were epitope mapped by surface plasmon resonance (SPR) using sensorchips displaying proteins within individual flow cell (Fc) channels (*top*). Binding is indicated by cyan and non-binding by purple (*bottom*). The geometric mean (geomean) $K_D$ (nM) for binders is indicated (see Methods for details and Supplementary Table 2 for SPR parameters). Only *Bicycles* with definitive target binding are indicated. **d** *Bicycles* from (**c**) were tested by AlphaScreen assay to determine whether they bound unique or overlapping epitopes (*top*). Combinations of epitope groups that compete for binding to Spike are indicated by cyan and those that don't in purple (*bottom*). **e** Modified AlphaScreen using RBD-binding *Bicycles* measuring the inhibition of binding between donor beads, coated with Spike, and acceptor beads, coated with ACE2 (*top*). A single example of each epitope group is shown (*bottom*). Dose-response curves were fit to obtain $IC_{50}$ values. Only E2 *Bicycles* inhibited, with a geometric mean $IC_{50}$ of 520 nM. Error bars represent the standard deviation relating two technical replicates. **f, g** Complexes of two different *Bicycles* belonging to epitope group 2 (E2) (PDB IDs: 7Z8O and 8AAA), superposed with an hACE2:RBD complex (https://www.rcsb.org/structure/6m0j). hACE2 is shown in blue, RBD in gray, and *Bicycles* in atom-colored spheres. **f** Complex with canonical E2 (green). **g** Complex with alternative E2, E2b (yellow). **h** RBD:E2b complex superposed with closed spike (https://www.rcsb.org/structure/6ZP0). **i** RBD:E2b complex superposed with open spike (https://www.rcsb.org/structure/7A98). Source data are provided as a Source Data file.

coated Nickel-chelate donor beads were incubated with Protein A acceptor beads displaying hACE2 as an Fc-fusion in the presence of different *Bicycles* (Fig. 1e). Amongst the different epitope groups, only E2 *Bicycles* were capable of directly competing for hACE2 binding (Fig. 1e and Supplementary Table 4, E2 $IC_{50}$ of 520 nM). When alternative sequences belonging to the same epitope group were used, they were differentiated by an additional letter (e.g., E2b or E2c). To

determine where on RBD E2 *Bicycles* bind, and the mechanism by which they compete with hACE2, the crystal structures of two distinct E2 peptides in complex with RBD were determined; E2 and E2b (Supplementary Table 5). As expected, both E2 *Bicycles* bind the same site within RBD; a narrow saddle-like surface adjacent to the hACE2 interface (Fig. 1f, g). Superposition with a RBD-hACE2 structure (6M0J)[39] suggests that E2 may be a direct hACE2 competitor (Fig. 1f) but this is

**Table 1 | Exemplar peptide sequences and consensus per epitope**

| Epitope | Scaffold | Sequence | | | | | | | | | | | | | | | |
|---|---|---|---|---|---|---|---|---|---|---|---|---|---|---|---|---|---|
| 1 | TATA | A | $C_i$ | E | Y | V | G | P | M | $C_{ii}$ | Y | R | L | Y | $C_{iii}$ | A | |
| | | A | $C_i$ | **E** | **Y** | x | **G** | **P** | x | $C_{ii}$ | x | **R** | **L** | **Y** | $C_{iii}$ | A | |
| 2 | TATA | A | $C_i$ | P | Y | V | A | G | R | G | T | $C_{ii}$ | L | L | $C_{iii}$ | A | |
| | | A | $C_i$ | **P** | x | x | x | **G** | x | **G** | x | $C_{ii}$ | **L** | **L** | $C_{iii}$ | A | |
| 2b | TATA | A | $C_i$ | M | F | V | P | $C_{ii}$ | A | V | R | H | A | L | G | L | $C_{iii}$ | A |
| | | A | $C_i$ | **M** | x | x | **P** | $C_{ii}$ | x | x | **R** | x | x | **L** | **G** | **L** | $C_{iii}$ | A |
| 3 | TATA | A | $C_i$ | E | D | N | D | W | V | Y | $C_{ii}$ | S | T | **$C_{iii}$** | A | |
| | | A | $C_i$ | x | x | x | x | **W** | x | **Y** | $C_{ii}$ | **S** | **T** | $C_{iii}$ | A | |
| 4 | TATB | A | $C_i$ | I | P | L | D | W | T | $C_{ii}$ | M | I | A | $C_{iii}$ | A | |
| | | A | $C_i$ | x | x | x | **D** | **W** | **T** | $C_{ii}$ | x | x | x | $C_{iii}$ | A | |
| 5 | TATB | A | $C_i$ | A | N | P | D | N | P | V | $C_{ii}$ | R | F | Y | $C_{iii}$ | A | |
| | | A | $C_i$ | x | **N** | x | x | **N** | **P** | **V** | $C_{ii}$ | **R** | **F** | **Y** | $C_{iii}$ | A | |
| 6 | TATB | A | $C_i$ | D | H | Y | H | $C_{ii}$ | P | W | L | A | L | G | G | S | $C_{iii}$ | A |
| 7 | TATB | A | $C_i$ | I | N | P | Y | $C_{ii}$ | E | H | H | I | Y | L | E | H | $C_{iii}$ | A |
| 9 | TATB | A | $C_i$ | M | N | P | F | F | Y | D | $C_{ii}$ | E | T | V | $C_{iii}$ | A | |
| | | A | $C_i$ | **M** | **N** | **P** | **F** | x | x | x | $C_{ii}$ | x | x | x | $C_{iii}$ | A | |
| 11 | TCMT | A | $C_i$ | F | P | E | P | W | L | G | L | $C_{ii}$ | T | P | $C_{iii}$ | A | |
| | | A | $C_i$ | **F** | **P** | x | **P** | **W** | **L** | **G** | **L** | $C_{ii}$ | **T** | **P** | $C_{iii}$ | A | |
| 12 | TATA | A | $C_i$ | S | S | K | F | $C_{ii}$ | D | A | W | W | N | F | N | R | $C_{iii}$ | A |
| | | A | $C_i$ | **S** | x | x | **F** | $C_{ii}$ | x | **A** | **W** | x | x | **F** | **N** | x | $C_{iii}$ | A |

The sequences of example peptides for each epitope group is shown together with the scaffold. Full name of scaffolds: TATA is 1,1′,1″-(1,3,5-triazinane-1,3,5-triyl)triprop-2-en-1-one, also known as triacryloylhexahydro-s-triazine; TATB is 1,3,5-tris(bromoacetyl) hexahydro-1,3,5-triazine; TCMT is 2,4,6-tris(chloromomethyl)-s-triazine.

less clear for E2b (Fig. 1g). Superposition of the E2b:RBD complex on the structure of full-length Spike in its 'closed' conformation (i.e., prior to hACE2 engagement; 6ZP0[40]), reveals that E2b binds between the RBD and NTD domains. Binding of hACE2 only occurs to the 'open' conformation of Spike (7A98[41]), in which RBD moves away from the NTD and adopts an upright orientation (Fig. 1i). It is, therefore, possible that E2b, in addition or instead of directly competing with hACE2, modulates Spike conformational flexibility, thus indirectly inhibits hACE2 engagement.

**Multimerisation and combinatorial assembly of bicyclic peptides generates potent antiviral inhibitors**

Two advantages of *Bicycles* are that they are highly modular and readily multimerized. This is particularly useful against oligomeric targets such as Spike, as it allows affinity enhancement through avidity (Fig. 2a). To exemplify this, we selected the E2 *Bicycle* and synthesized multimeric variants using an amido-PEG10-triazolyl linker with appropriate valency hinge (Glutaric Acid (GTA), Tri(carboxyethyloxyethyl)amine (TCA), and 1,3-bis(carboxyethoxy)-2,2-bis(carboxyethoxy)propane (TET)). We measured how multimerization altered the degree of Spike-hACE2 competition using our AlphaScreen (Figs. 1e, 2b and Supplementary Table 6). In parallel, we assayed whether a multimerized E2 was capable of inhibiting infection (Fig. 2b and Supplementary Table 7) using an S-pseudotyped (S-pV) lentiviral system, in which S-displaying virions infect 293 T cells overexpressing hACE2 and TMPRSS2[42]. Multimerizing E2, from monomer through to tetramer, concomitantly improved both the affinity to Spike and the potency with which they were able to inhibit infection (Fig. 2b and Supplementary Tables 6 and 7). Dimerization improved the $IC_{50}$ for infection inhibition > 20-fold to sub-μM, while trimerization and tetramerization improved the $IC_{50}$ > 4,000-fold ($IC_{50}$ values: monomer = 670 nM, dimer = 92 nM, trimer = 0.086 nM, tetramer 0.096 nM). Notably, similar Spike affinities and $IC_{50}$ were observed for trimer and tetramer E2 *Bicycles*. This is consistent with Spike being a trimeric protein and that multimerized *Bicycles* bind to multiple Spike monomers within each trimer, rather than to Spike monomers across separate Spike trimers. To further explore this, we made a series of E2 trimers with different TCA-amido-PEG-triazolyl monomer linker lengths ranging from PEG1 to PEG23 (approx. 15 to 95 Å). Near identical nanomolar $IC_{50}$ values were obtained for all linker lengths, including the shortest linker, PEG1 which is not long enough to permit binding across multiple spike trimers (Fig. 2a, c, Supplementary Table 7; PEG1 = 5.8 nM, PEG5 = 7.3 nM, PEG10 = 8.5 nM and PEG23 = 5.6 nM). Together with the AlphaScreen competition data, this suggests that the mechanism of infection inhibition is through competition with hACE2. Importantly, these results show that *Bicycles* as a drug modality enable hits to be rapidly turned into potent nanomolar antivirals without the laborious iterative structure activity relationship driven chemical improvements normally required during preclinical drug optimization.

We investigated the activity of multimerized binders that, as monomers, did not directly compete with hACE2 and made trimeric molecules from an E3 *Bicycle* and an E5 *Bicycle*. In each case, the E3 and E5 trimerized *Bicycles* inhibited S-pV infection, with potencies of $IC_{50}$ 150 nM and 82 nM respectively (Fig. 2d and Supplementary Table 7) whilst the monomer versions showed little or no activity. E3 binders bind to a separate epitope and are non-competitive with E2, illustrating that antivirals can be made from molecules that bind different and non-overlapping parts of the RBD surface. How E3 and E5 multimerized *Bicycles* inhibit infection is unclear, given the inability of their parent monomers to compete for hACE2 binding. Conceivably, when every monomer on Spike is engaged by a *Bicycle*, binding to hACE2 may be sterically prevented, perhaps via the PEG linkers that connect each *Bicycle* monomer. Alternatively, multimerized E3 and E5 might bind to Spike hinge regions, preventing the necessary conformational changes that allow ACE2 binding. Importantly, and independent of the exact mechanism of inhibition, these results demonstrate that effective antivirals can be made from non-competitive precursors and highlight the advantage of screening for binders to Spike rather than for inhibitors of ACE2 interaction.

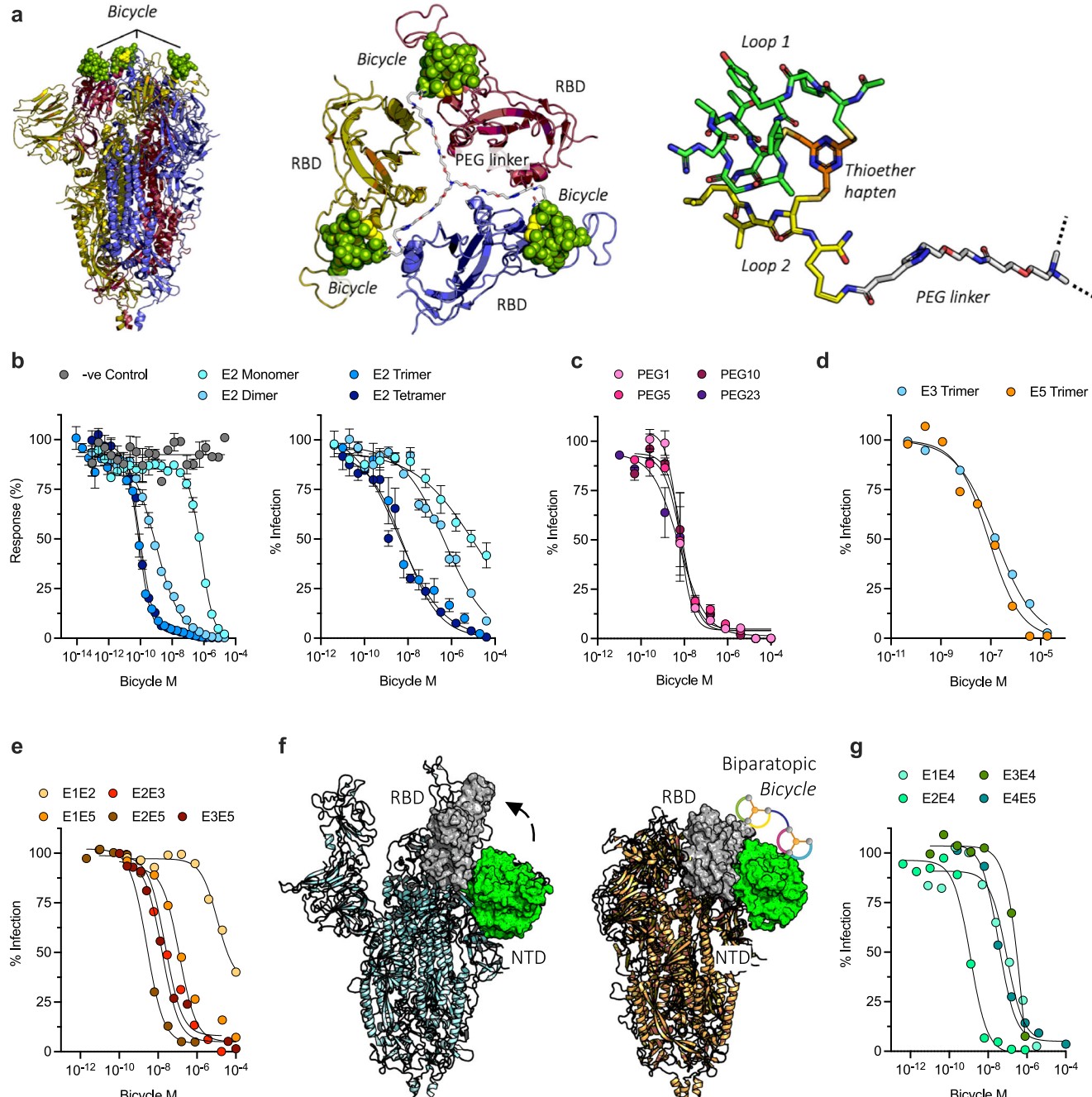

**Fig. 2 | Multimerisation and combinatorial assembly generates potent inhibitors. a** E2:RBD complex superposed on closed spike (PDB ID:6ZP0). Top-down view showing E2 *Bicycle* and RBDs from each Spike monomer (red, yellow, blue). A TCA-amido-Peg1-triazolyl linker is modeled connecting the three E2 *Bicycle* molecules. An individual E2 *Bicycle* is also shown, indicating the constraining hapten, peptide loops and PEG linker. Dotted lines indicate where other *Bicycle* monomers are attached via a TCA hinge. **b** E2 was multimerized into dimer, trimer and tetramer versions using an amido-PEG10-triazolyl linker with appropriate valency. Multimerised E2 *Bicycles* inhibition of Spike binding to hACE2 (left-hand graph) or Spike-pseudotyped virus (S-pV) infection of 293 T TMPRSS2/hACE2 cells (right-hand graph). Dose-response or infection curves were fit to obtain $IC_{50}$ values. **c** E2 was trimerized with increasing TCA-amido-PEG-triazolyl linker lengths from PEG1 to PEG23. Dose-response curves for S-pV infection inhibition were fit to obtain $IC_{50}$ values. **d** Other Spike binding, but non-ACE2 competing, *Bicycles* were trimerized. E3 trimer utilized a TCA-amido-PEG23-triazolyl linker and E5 trimers utilized a N-

(acid-PEG3)-N-bis(PEG3-amido)-PEG36-triazolyl linker where the central hinge conjugation point was not extended by PEG36. Dose-response curves for S-pV infection inhibition were fit to obtain $IC_{50}$ values: E3 = 150 nM, and E5 = 82 nM. **e** Dose-response curves for S-pV infection inhibition by biparatopic *Bicycles* utilizing a non-ACE2-competing RBD molecule. The following $IC_{50}$ values were obtained: E1E2 = 11000 nM, E2E3 = 23 nM, E2E5 = 2.6 nM, E1E5 = 120 nM, and E3E5 = 16 nM. **f** Models of Spike in its 'open' (7A98) and 'closed' conformations (6ZP0), with the NTD and RBD domains shown as green and gray surfaces respectively. An arrow indicates that the RBD dissociates from the NTD in the open state. The expected binding of the biparatopic *Bicycle* E2E4 to both RBD and NTD domains is shown. **g** Dose-response curves for S-pV infection inhibition by biparatopic *Bicycles* utilizing an NTD-binding molecule. In infection experiments, the data is the result of two independent experiments presented as mean values (**d**, **e**, **g**) or three independent experiments as mean ± SEM (B&C). Source data are provided as a Source Data file.

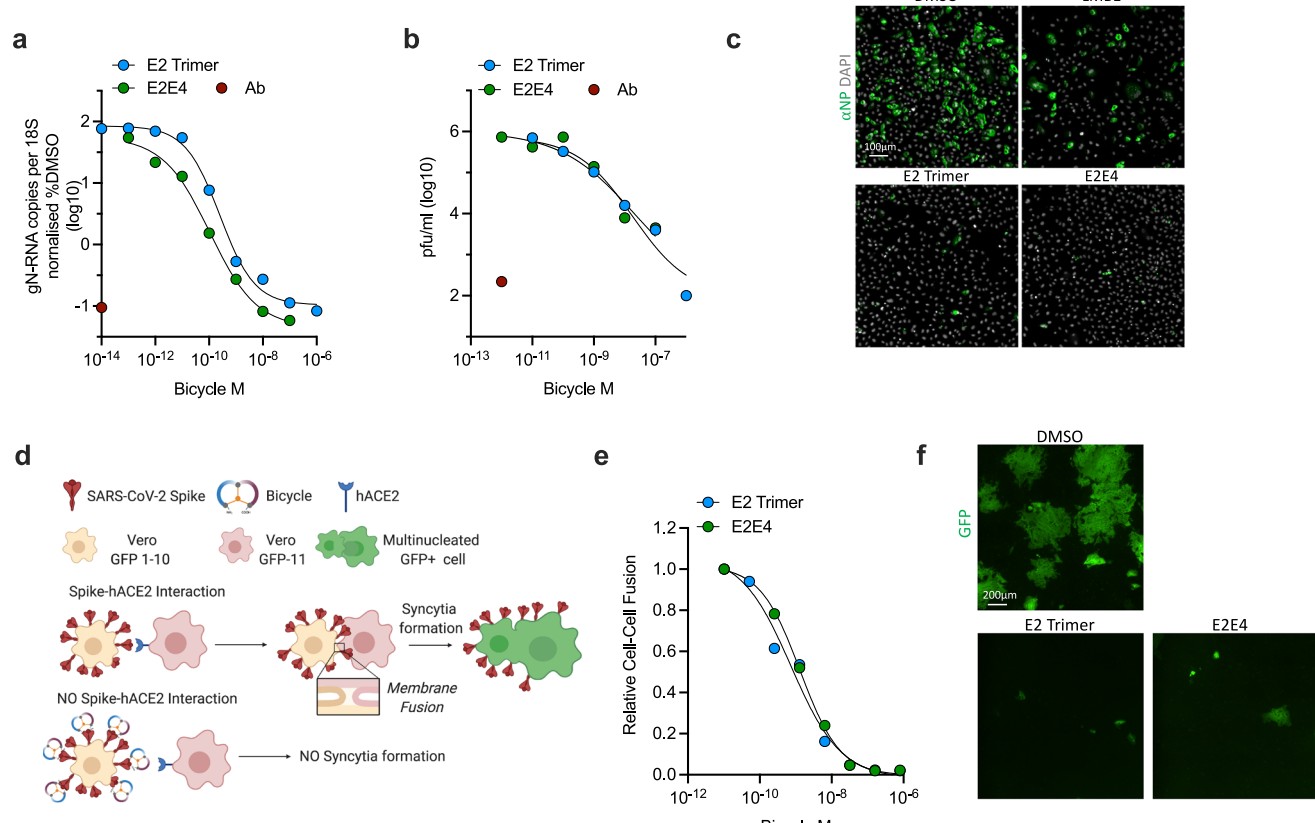

**Fig. 3 | *Bicycles* inhibit SARS-CoV-2 replication and cell–cell fusion. a–c** SARS-CoV-2 (Wuhan-Hu-1 strain) was used to infect Vero TMPRSS2/ACE2 cells in the presence of homotrimeric E2 or biparatopic E2E4. Replication was measured after 18 hours by qPCR for gN-RNA (**a**), 3 days by plaque assay (**b**) or cells were fixed and stained with anti-NP antibody (antibodies-online, ABIN6953059) after 18 hours (**c**). Polyclonal convalescent serum (LMB1) was used as a positive control. Data is the result of two independent experiments presented as mean values. Dose response curves were fitted to obtain qPCR $IC_{50}$ values of: E2 trimer = 0.25 nM, E2E4 = 0.11 nM and plaque assay $IC_{50}$ values of: E2 trimer = 1.8 nM and E2E4 = 2.0 nM. The limit of detection for plaque assays was $\log_{10}(0.82)$ pfu/ml. **d** Schematic for cell–cell fusion assay. Two stable Vero cell lines each expressing one half of split GFP and either Spike or hACE2 were mixed and the formation of multinucleated syncytia

monitored by the resulting GFP signal using an IncuCyte S3 live microscope. Created with Biorender.com. **e** Dose response curves for the inhibition of GFP+ multinucleated syncytia formation by E2 trimer and E2E4 biparatopic *Bicycles*, based on immunofluorescence data shown in Supplementary Figure 1B. Data is plotted as relative cell–cell fusion, where fusion in DMSO in the absence of any *Bicycle*, ( ∼ 30% of cells), is set at 1. Fitting the data results in $IC_{50}$ values of: E2 trimer = 0.74 nM and E2E4 = 1.3 nM. Data shown is from two independent experiments presented as mean values. **f** Immunofluorescence images of GFP+ multinucleated syncytia formed in the presence of DMSO, 32 nM E2 trimer or 32 nM E2E4 *Bicycles*. Data is representative of two independent experiments. Source data are provided as a Source Data file.

*Bicycles* do not need to be multimerized using the same monomer. Instead, different *Bicycles* can be combined to make biparatopic molecules that simultaneously bind two different epitopes. Combining two different non-competitive Spike binders together generated biparatopics capable of inhibiting infection with a range of activities, from micromolar to nanomolar $IC_{50}$ (Fig. 2e and Supplementary Table 7). This biparatopic approach also enables molecules to be designed that simultaneously engage and bind to two different domains on Spike which potentially allows additional inhibitory mechanisms to be generated that go beyond simple hACE2 competition. For instance, Spike undergoes multiple conformational rearrangements during viral entry, such as movement of RBD with respect to NTD, and preventing these could also block infection. To test this, we synthesized a biparatopic to simultaneously bind both RBD and NTD, reasoning that this may constrain RBD movement into the open conformation and therefore interfere with receptor engagement (Fig. 2f). This generated antivirals with micromolar to nanomolar potency (Fig. 2g and Supplementary Table 7; E2E4 = 1.2 nM, E1E4 = 560 nM, E4E5 = 30 nM, E3E4 = 680 nM). This illustrates the utility of the large number of unique epitope-binding *Bicycles*, which were obtained by unbiased target panning, as a toolbox of molecules. These can be used

combinatorially to build innovative molecules which bind multiple parts of Spike simultaneously to access novel inhibitory mechanisms (Fig. 1b).

## *Bicycles* inhibit SARS-CoV-2 replication and cell–cell fusion

We selected an exemplar homotrimer (E2 trimer) and biparatopic (E2E4) *Bicycle* and tested whether they could limit the replication of live SARS-CoV-2 virus. We used a clinical isolate of SARS-CoV-2/human/Liverpool/REMRQ0001/2020 and Vero cells modified to stably express hACE2 and TMPRSS2[42]. Cells were infected in the presence of either E2 trimer or E2E4 and replication assessed after 24 hours by measuring viral genome synthesis, viral protein expression and the production of infectious particles. Data from qPCR using probes against genomic N and plaque assays of budded virions revealed that both the homotrimeric and biparatopic *Bicycles* were capable of reducing replication to similar levels as a potent neutralizing antibody control (Fig. 3a, b and Supplementary Table 7). The calculated $IC_{50}$ values for N transcription inhibition were 0.25 nM for E2 trimer and 0.11 nM for E2E4, and 1.8 nM for E2 trimer and 2.0 nM for E2E4 when inhibiting viral production, illustrating that *Bicycles* suppress SARS-CoV-2 replication with similar potency to S-pV neutralization. Immuno-histochemistry on infected

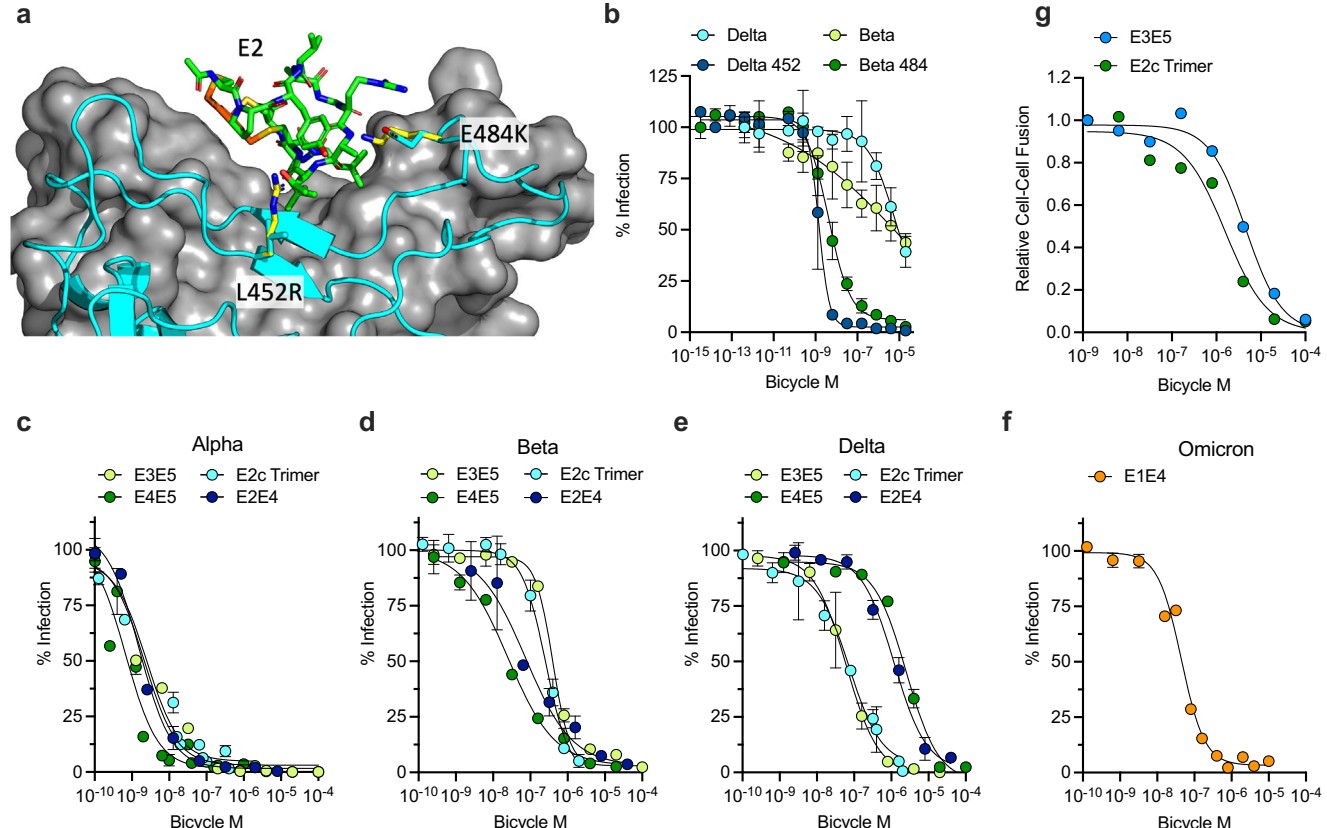

**Fig. 4 | Multimeric and biparatopic *Bicycles* inhibit SARS-CoV-2 VoC. a** Structure of E2 *Bicycle* in complex with ACE2 showing location of two key VoC mutations L452R (in Delta) and E484K (in Beta). **b** Infection of 293 T TMPRSS2/ACE2 cells with S-pV displaying the Spike protein from Delta or Beta VoCs, or point mutations L452R (Delta 452) or E484K (Beta 484), in the presence of E2 trimer. Data is the result of three independent experiments presented as mean values +/- SEM. Dose response curves were fitted to obtain $IC_{50}$ values: Delta = 5700 nM, Delta 452 = 1.4 nM, Beta = 19000 nM, Beta 484 = 4.2 nM. **c–f** Infection of 293 T TMPRSS2/ ACE2 cells with S-pV displaying the Spike protein from VoCs. Data is the result of three independent experiments presented as mean values +/- SEM. Dose response

curves were fitted to obtain $IC_{50}$ values for Alpha (**c**): E3E5 = 2.1 nM, E4E5 = 0.67 nM, E2c Trimer = 2.5 nM, E2E4 = 1.6 nM. For Beta (**d**): E3E5 = 400 nM, E4E5 = 25 nM, E2c Trimer = 250 nM, E2E4 = 74 nM. For Delta (**e**): E3E5 = 61 nM, E4E5 = 2500 nM, E2c Trimer = 73 nM, E2E4 = 1200 nM. For Omicron (**f**): E1E4 = 44 nM. **G** Dose response curves for the inhibition of GFP + multinucleated syncytia formation by cells expressing Delta Spike by an epitope group 2 *Bicycle* E2c or E3E5 biparatopic *Bicycle*, fit to obtain $IC_{50}$ values of: E2c trimer = 1.6 μM and E3E5 = 4.5 μM. Data shown is from two independent experiments presented as mean values. Source data are provided as a Source Data file.

cells probing for N-protein expression further supported that *Bicycles* potently inhibited infection (Fig. 3c and Supplementary Fig. 1A)[42]. To assess whether *Bicycles* can also block this mode of viral spread, we utilized a split-GFP cell fusion assay, in which cells expressing one part of GFP and Spike were co-cultured with cells expressing the second part of GFP and hACE2[43]. Cell fusion can then be monitored by real time measurement of GFP fluorescence at different *Bicycle* concentrations (Supplementary Fig. 1B, C). Both the homotrimeric and biparatopic *Bicycles* were capable of inhibiting Spike-mediated syncytia formation (Fig. 3e, f and Supplementary Table 7), with similar nanomolar potencies as for cell-free infection (E2 trimer = 0.74 nM and E2E4 = 1.3 nM).

### *Bicycles* selected against Wuhan-Hu-1 Spike are active against other SARS-CoV-2 variants

In the short period since the start of the COVID-19 pandemic, multiple SARS-CoV-2 variants of concern (VoC) have emerged. Each VoC has multiple mutations in Spike with respect to the original Wuhan-Hu-1 strain that alter tropism, reduce vaccine efficacy and evade neutralizing antibodies. Structural analysis of the E2 used in both the homotrimeric E2 and biparatopic E2E4 reveals that binding takes place close to two positions on the RBD that vary in both the Beta and Delta VoC (Fig. 4a). Testing the E2 trimer in S-pV assays using either Beta or Delta Spike shows that inhibition potency is substantially reduced (Fig. 2b

and Supplementary Table 7). However, potency is restored when positions 452 or 484 are returned to the original Wuhan-Hu-1 residues (i.e., L452R- > L452 or E484K- > E484) demonstrating that Spike mutants in VoC can impact *Bicycle* efficacy (Fig. 2b and Supplementary Table 7). However, because *Bicycles* were generated to many different epitopes across Spike (Fig. 1b), we reasoned that by selecting *Bicycle* molecules that bound to less variant regions and by combining these into new homotrimeric or biparatopic formats we could rapidly generate new inhibitors against VoCs without needing to re-screen for additional binders. We tested this hypothesis by screening a small panel of biparatopic and homotrimeric *Bicycles* against Alpha, Beta, Delta and Omicron VoC. In each case, we were able to identify a *Bicycle* combination that inhibited each VoC (Fig. 4c–f and Supplementary Table 7). This approach also generated inhibitors capable of blocking cell–cell fusion mediated by a Delta VoC Spike. In cells expressing Delta Spike, the same homotrimer and biparatopic *Bicycles* that blocked infection also inhibited syncytia formation (Fig. 4g and Supplementary Table 7).

### *Bicycles* inhibit SARS-CoV-2 replication in both mouse and hamster in vivo models

Encouraged by the potent inhibitory activity of anti-Spike *Bicycles* in S-pV, cell–cell and live virus assays, we tested whether they could be used to treat infection in animal models. Both *Bicycles'*

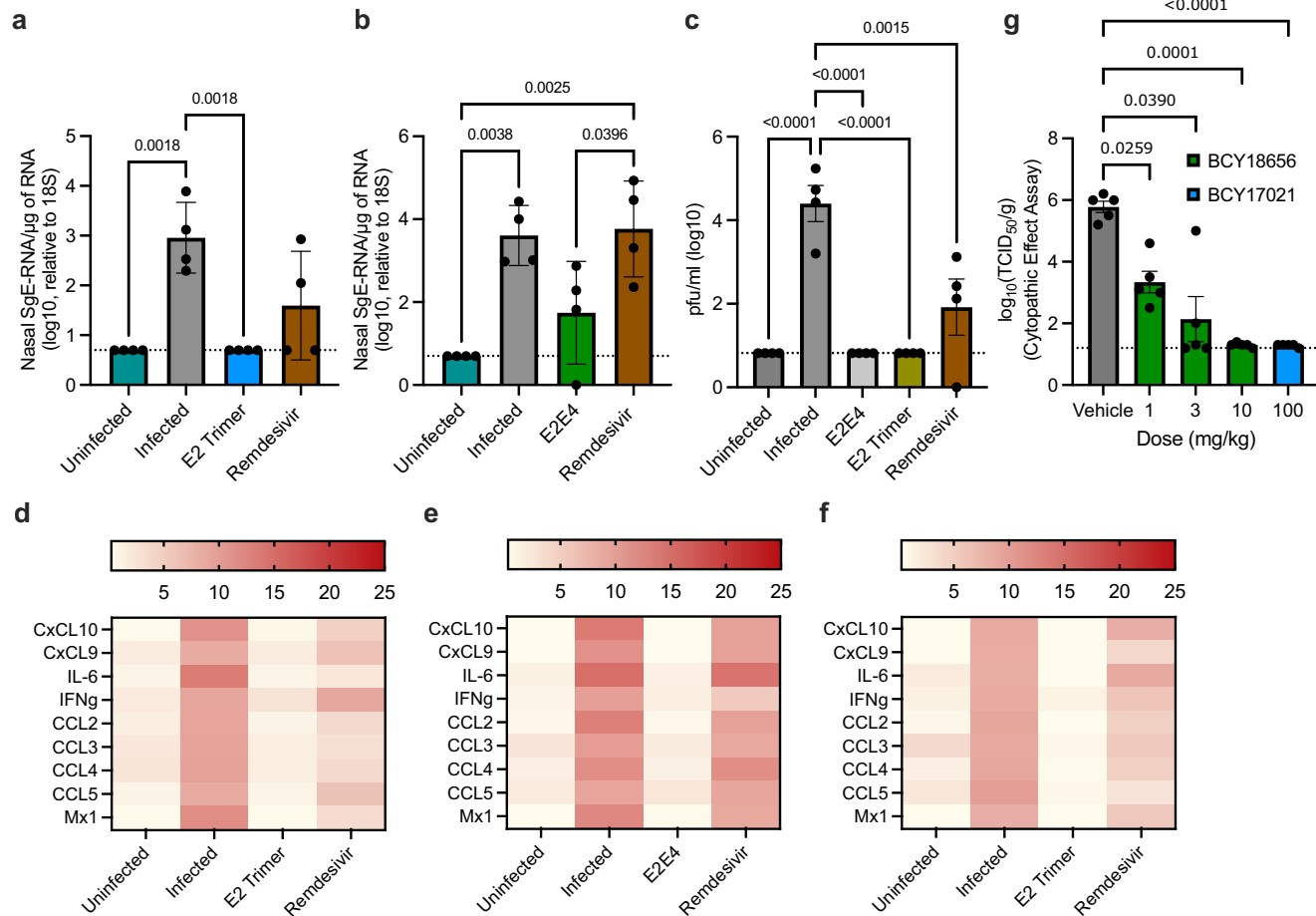

**Fig. 5 | *Bicycles* inhibit SARS-CoV-2 replication in vivo. a–f** K18 hACE2-mice (N = 4) were given *Bicycle* (900 mg/kg/day), Remdesivir (50 mg/kg/day) or control subcutaneously for five days, initiated 24 h prior to infection. SARS-CoV-2 (Liverpool/REMRQ0001/2020) was administered intranasally on day 0 at $10^4$ PFU/mouse or a vehicle-only control. On day 4, mice were culled and lungs and nasal turbinates removed. **a, b** Viral replication was measured by qPCR using probes against viral SgE. Each data point corresponds to one mouse. Data are presented as mean values +/- SEM. The limit of detection was $\log_{10}(0.7)$ SgE/μg RNA. **c** Active viraemia in lungs was measured by plaque assay on Vero TMPRSS2/ACE2 cells of lung homogenate. The limit of detection for plaque assays was $\log_{10}(0.82)$ pfu/ml. Data are presented as mean values +/- SEM. Cytokine transcription was measured by qPCR in nasal turbinate (**d**) or lung homogenate (**e, f**) using a panel of probe sequences, as indicated, and expressed as a fold-change over K18 hACE2-mice controls. **g** Syrian hamsters (N = 6) were given *Bicycle* or vehicle control subcutaneously three times daily at the indicated doses for five days. SARS-CoV-2 (BetaCoV/Munich/BavPat1/2020) was administered intranasally on day 2 at $10^2$ $TCID_{50}$/mouse. On day 5, mice were culled and viraemia measured in lung homogenate by cytopathic assay on Vero E6 cells. The limit of detection ranged between 1.2 to 1.4 log10 (TCID50/g). Data are presented as mean values +/- SEM. For all data, non-parametric one-sided ANOVA was used for statistical analysis and only significant differences indicated. Data with a value of zero was set to the limit of detection for statistical purposes. Source data are provided as a Source Data file.

pharmacokinetic parameters were evaluated in CD-1 mice and Syrian golden hamsters (Supplementary Fig. 2A–D). We identified that subcutaneous administration gave extended systemic half-life over intravenous infusion in all cases as a more durable profile for antiviral activity (Supplementary Fig. 2E). Using pharmacokinetic-pharmacodynamic (PKPD) models constructed with the antiviral and pharmacokinetic datasets, we predicted target coverage over time (Supplementary Fig. 3A–D). This indicated the compounds would be most suited to a *ter in die* (t.i.d.) dosing regimen to maintain 90% viral coverage at trough. The PKPD models also elucidated the dose that gave most confidence, in respective species for each compound, for efficacy according to the t.i.d regimen. We infected K18 hACE2-transgenic mice intranasally with SARS-CoV-2/human/Liverpool/REMRQ0001/2020 at $10^4$ PFU/mouse and treated animals prophylactically with either E2 trimer or E2E4 *Bicycle*, or remdesivir as a positive control. Treatment was given subcutaneously either three times per 24 h for *Bicycles*, or twice per 24 h for remdesivir, starting one day prior to infection. Four days post-infection, animals were culled and lungs and nasal turbinates tested for evidence of viral

replication. Treatment with either E2 or E2E4 *Bicycles* significantly reduced viral RNA levels in nasal turbinates, as measured by qPCR using probes against the sub-genomic viral E gene, in contrast to remdesivir which had no significant impact (Fig. 5a, b). We were unfortunately unable to perform plaque assays on nasal turbinates due to challenges in processing and shipping samples between labs during the pandemic. However, we were able to determine whether *Bicycle* treatment reduced the levels of infectious virus in the lung by performing a series of plaque assays on lung homogenate. The plaque data revealed that in animals treated with either E2 trimer or E2E4, there was no detectable infectious virus in the lungs on day 4 postinfection (Fig. 5c). In contrast, there were considerable levels of infectious virus in the lungs of untreated mice whilst remdesivir only partially decreased viraemia. The clinical pathology of COVID-19 is associated with a hyperinflammatory state, and in severe cases this escalates to Acute Respiratory Distress Syndrome (ARDS). A panel of cytokine and chemokine transcripts was also measured in lung and nasal turbinate homogenate from the in vivo models to determine whether *Bicycles* could reduce inflammation concomitantly with virus

clearance. Infection clearly resulted in a marked increase in multiple inflammatory markers which remained substantially elevated in remdesivir-treated animals and untreated controls (Fig. 5d–f). In contrast, animals given either E2 homotrimer or E2E4 biparatopic *Bicycles* showed an inflammatory signature similar to that in uninfected mice.

An alternative rodent model for SARS-CoV-2 infection is the Syrian golden hamster, in which the virus uses endogenous rather than transgenic human ACE2 as the cellular receptor. The challenge experiments were repeated using the hamster model but with a lower viral inoculum of $10^2$ $TCID_{50}$ SARS-CoV-2. As before, *Bicycles* were dosed prophylactically subcutaneously three times per 24 h, starting one day prior to infection. Hamsters were given either 1, 3, or 10 mg/kg per dose of E2E4 biparatopic *Bicycle* or 100 mg/kg of E2 homotrimer. On day 4 postinfection, lung homogenate was tested for the presence of infectious virus by cytopathic assay. Treatment with either *Bicycle* suppressed viral replication (Fig. 5g). Viral inhibition was dose-dependent, with 10 mg/kg dosing using the biparatopic *Bicycle* sufficient to reduce viraemia to undetectable levels (Fig. 5g). Taken together, both rodent models confirm the virological efficacy of homotrimeric and biparatopic *Bicycles* as SARS-CoV-2 antivirals.

## Discussion

Here we show that a novel drug modality, called *Bicycles*, allows the rapid development of potent SARS-CoV-2 antivirals. Antiviral *Bicycles* are effective against diverse VoC and prevent infection and inflammation in animal models. Screening a phage library of randomized *Bicycle* sequences against viral Spike generated a large number of hits across the entire surface of the protein. These initial hits were rapidly progressed to potent inhibitors simply by joining multiple *Bicycles* together using PEG linkers. This allowed the creation of homomultimers, such as homotrimers that bind simultaneously to all three monomers on a Spike trimer and block SARS-CoV-2 infection. This also allowed the creation of biparatopics; molecules that combine *Bicycles* targeting different epitopes, or even different domains, and which have potent antiviral activity. Importantly, even *Bicycles* that individually are unable to compete for hACE2 receptor binding and lack antiviral activity can be combined in either multimer or biparatopic formats to make effective inhibitors. This ability to generate multiple initial hits, characterize them into distinct epitope groups and then combine them in different formats means that each selection generates not a single inhibitor but a toolbox of molecules that can be rapidly mixed and matched together to make many different bespoke antivirals. This allows the targeting of different parts of the viral protein to avoid regions that are mutated in VoC. We have illustrated this by generating potent antivirals against diverse VoCs simply by combining together multiple *Bicycles* generated against Wuhan-Hu-1 Spike protein. Our toolbox approach also allows us to access different inhibitory mechanisms, not just receptor blocking. For instance, we show that a potent antiviral can be made using a biparatopic that links together epitopes from different parts of Spike protein, potentially impacting on conformational changes that are required to promote viral entry. Finally, we show that homotrimer and biparatopic *Bicycles* inhibit SARS-CoV-2 infection in both hACE2-transgenic mice and Syrian golden hamster models of SARS-CoV-2 infection, and prevent a hyperinflammatory state, a condition associated with COVID-19 pathology. Using animal models globally accepted as benchmarks for viral efficacy such as the ACE2 transgenic mouse or Golden Syrian Hamster[44], antibodies and other biologic agents targeting the spike protein of SARS-CoV-2 have historically been shown to block the ability of virus to enter host cells and prevent or treat viral pathology[45,46]. Many of the agents effective in such animal models were subsequently approved for use in humans and shown to result in extended protection against severe infection and reduction in hospitalization and deaths[5,8]. Such protective efficacy was usually only reduced by the impact of viral mutations in key epitope residues[47,48].

As *Bicycles* were delivered thrice daily to maintain provisional target exposures, the presented data should not be interpreted as postulating future medicines requiring thrice daily dosing in humans. Improving metabolic stability for longer durations of exposure and formulations for different routes of administration are typical next steps to address posological challenges. The design principles of our multimeric *Bicycles* are broadly similar to that of the clinical biologic, Ensovibep[45,49,50]. Ensovibep encompasses three distinct SARS-CoV-2 RBD-binding Designed Ankyrin Repeat Proteins (DARPins) and two distinct human serum albumin (HSA)-binding DARPins. Ensovibep demonstrated anti-SARS-CoV-2 efficacy in both prophylactic and therapeutic preclinical in vivo models[45,49]. This compound has recently completed early clinical evaluation and has shown therapeutic benefit in preventing infection, warranting further Phase II evaluation[51]. This data is encouraging for the use of non-small molecule, non-antibody based therapies and suggests that pharmacokinetics need not be an obstacle to use in humans. However, the in vivo pharmacokinetic data presented in the current manuscript for *Bicycles* was generated to support the in vivo study design and interpretation of the results rather than predict human pharmacokinetics and simulation of the human plasma concentration-time profile. Future work will be required in other species, including non-human primates, which have previously been shown to provide reliable estimates of human pharmacokinetic parameters for peptide-based therapeutics[52].

Further experiments will also be required to explore treatment efficacy beyond 4 days. However, we are encouraged by the fact that by 4 days viraemia was undetectable in some conditions suggesting that sterile immunity had been achieved. The correlation between viral clearance and suppression of an inflammatory response we observed is also encouraging for positive patient outcomes. Moreover, this correlation, combined with the ready availability of diagnostic tests for SARS-CoV-2, suggests that viraemia represents a viable clinical and therapeutic endpoint. All viruses must interact with surface receptors to infect cells and the ability of neutralizing antibodies to prevent infection by inhibiting this interaction demonstrates the applicability of such an antiviral mechanism to diverse viruses of public health importance. Given that this is also one mechanism by which the *Bicycles* described here prevent SARS-CoV-2 infection, it seems reasonable to suppose that *Bicycles* developed against the cell-surface receptor binding proteins of other viruses would be useful to treat other viral infections. DARPins have been successfully developed to inhibit HIV-1[53] infection as well as SARS-CoV-2[50] and have been proposed as broad antivirals on the basis of inhibiting receptor binding[49]. One potential benefit of *Bicycles*, as illustrated here, is the speed with which specific antiviral molecules can be generated. This property makes *Bicycles* ideal as an antiviral modality in the event of future pandemics caused by new or newly zoonosed viruses.

Based on the results here, we suggest that *Bicycles* – chemically-constrained bicyclic peptides that can be displayed in massive phage libraries – offer a new potential format for antiviral drugs. They can be generated quickly, do not require laborious iteration to generate potent inhibitors and can be combined to access new inhibitory mechanisms and pre-empt resistance. All viruses must engage with cellular receptors to mediate entry and most use oligomeric, often trimeric, receptor binding proteins that are ideal targets for multimerized *Bicycles*. Receptor binding proteins are also typically conformationally plastic in order to promote membrane fusion, making them particularly sensitive to multi-paratopic drugs, such as those described here, that constrain different domains together. Thus, viruses are ideal targets for antiviral *Bicycles* and our combinatorial approach. Future work will seek to develop *Bicycles* against other respiratory viruses, such as influenza, and other diverse pathogens.

## Methods

### Selection of SARS-CoV-2 spike protein specific *Bicycles* by Phage Display

*Bicycle* bacteriophage (phage) libraries, consisting of linear peptides (10 to 20 amino acids in length) containing 3 cysteines which are cyclized in situ to form thioether-bonded bicyclic peptide libraries were used in selections against either S1 (ACROBiosystems S1N-C82E8), S1-RBD (ACROBiosystems SPD-C82E9), S2 (ACROBiosystems, S2N-C52E8) or Spike Trimer (pCOV55) recombinant proteins. Four rounds of selection were performed, using decreasing target concentrations of protein immobilized onto streptavidin magnetic beads. Binders were eluted at low pH. After round 4, phage clones were isolated, sanger sequenced, and tested for binding to proteins using *AlphaScreen* assay. Binders were chemically synthesized as peptides for further characterization. Certain lead clones subsequently returned to the phage platform for further affinity maturation. For these, custom *Bicycle* phage libraries were constructed based on the initial sequence, retaining residues thought to be important for binding and randomizing others. Phage selections using these custom libraries were carried out as before to identify binders with improved affinity over the parent sequences.

### RBD protein expression

Plasmid encoding for RBD as a His-Zbasic fusion construct (pExp-His-Zbasic-RBD, https://www.addgene.org/194999/) were transformed into BL21(DE3) cells and grown overnight on ampicillin plates. Single colonies of the cells were grown in 2xYT media at 37 °C until O.D. at 600 nm reached 0.6 then expression was induced with 0.4 mM IPTG. The cells were incubated for 5 hours at 37 °C then harvested by centrifugation at $4200 \times g$ for 25 minutes. The cell pellet was resuspended in 40 ml of 20 mM Tris-HCl pH 8.0, 100 mM NaCl and frozen at -80 °C. The cell suspension was thawed and 250 μl of 2 mg/ml DNase and 500 μl 100 mM PMSF added. The cells were lysed using a high-pressure homogenizer. The crude cell lysate was centrifuged for 20 min at $20,000 \times g$ (+4 °C) and the supernatant discarded. The cell pellet was resuspended in 25 ml of 20 mM Tris-HCl pH 8.0, 1% Triton X-100, 10 mM EDTA, 1 mM TCEP, and centrifuged for 10 min at $15,000 \times g$ (+4 °C). The cell pellet was resuspended in 25 ml of 20 mM Tris-HCl pH 8.0, 1000 mM NaCl, 1 mM TCEP, and centrifuged for 10 min at $15,000 \times g$ (+4 °C). The pellet was resuspended in 5 ml of 80 mM Tris-HCl pH 8.0, 800 mM NaCl, 4 mM TCEP. 15 ml 8 M guanidine-HCl was added. Suspension was centrifuged for 20 min at $40,000 \times g$ at +4 °C. The supernatant was loaded onto 5 ml of Ni-NTA resin, washed with 50 ml of 20 mM Tris-HCl pH 8.0, 500 mM NaCl, 10 mM imidazole, 6 M urea (deionised), 0.5 mM TCEP. The protein was eluted with 25 ml of 20 mM Tris-HCl pH 8.0, 100 mM NaCl, 200 mM imidazole, 6 M urea (deionised), 0.5 mM TCEP. The sampled was cooled down to =+4 °C. 500 ml of refolding solution (100 mM Tris, 500 mM Arg-HCl, 0.5 g/l PEG 3350, 1 M urea (deionised), 2 mM cysteine, 0.2 mM cystine) was cooled to +4 °C. Eluate from the Ni column was added into refolding solution while being stirred at 4 °C. Refolding was allowed to proceed for 24-72 h. The refolding solution was diluted 2× by adding 500 ml of 50 mM MES pH 6.0. Solution was filtered through a Whatman GF/B microfiber filter. Protein solution was loaded onto a 5 ml HiTrap SP HP column (Cytiva). Column was washed with 5 CV of 20 mM HEPES-NaOH pH 7.2, 50 mM NaCl. Protein was eluted with a 25 CV gradient 0-100% of 20 mM HEPES-NaOH pH 7.2, 1000 mM NaCl. RBD-containing fractions were pooled and 100 μl of 1 mg/ml His-TEVpro protease was added. Cleavage proceeded with rotation at +4 °C. Protein solution was passed through an equilibrated Ni-NTA resin to remove fusion tag and His-TEVpro. (Ni-NTA resin equilibrated in 20 mM HEPES-NaOH pH 7.2, 50 mM NaCl, 20 mM Imidazole.) Protein was concentrated to 1 ml using Amicon Ultra concentrator with cut-off 10 K. Protein was loaded onto a HiLoad Superdex 75 16/600 pg column and size exclusion chromatography performed in 20 mM Tris pH 7.2, 200 mM NaCl. Peak

fractions were pooled and protein concentrated to 15–16 mg/ml using Amicon Ultra concentrator with cut-off 10 K.

### Spike protein expression

Stable forms of the Spike protein trimer (S-R/PP, used in epitope binning; S-GSAS/PP, used in SPR and spike-ACE2 competition assays; S-R/x2, used in panning for the identification of some N-terminal epitope binding *Bicycles*)[40], were expressed in Expi293 cells provided by a previous study[54]. Protein was expressed in Expi293 cells, prepared in the following manner: Cells were cultured in Expi293 medium in 37 °C. The day before lentiviral infection, cells were moved from suspension into 20 mL DMEM media with 10% FBS in T175 flasks at a final density of $3×10^6$ cells/flask. On the day of infection, 20 ml of the virus-containing supernatant was warmed up to room temperature before adding to the cells. Cells were cultured for 24 h before adding 20 ml of fresh DMEM and 10% FBS, and monitored daily. Once recovered, they were moved to suspension. To this end, the media was removed and cells washed 3 times with PBS, before adding trypsin. Trypsin reaction was neutralized with 20 ml DMEM and 10% FBS and centrifuged for 3 min. The cells were diluted to a final density of $0.8×10^6$ cells/ml and cultured in suspension in 37 °C. For protein production, the culture was gradually expanded to 1.5 L at $2-3×10^6$ cells/ml, and transferred into 33 °C. Cells were harvested 6 days later from the cell culture supernatant by centrifugation at 180 x g and buffered with 10xPBS to a final concertation of 1x. Protein purification was performed as described previously[40]. 2.5 mM imidazole, 300 mM NaCl and 0.5 mM phenylmethylsulfonyl fluoride were added. The supernatant was loaded into a 5 ml Talon Cobalt (Takara Bio) column 2 times, followed by a 50 ml wash with 25 mM phosphate pH 8.0, 300 mM NaCl, 5 mM imidazole. Protein was eluted with 100 ml linear gradient to 100% of 25 mM phosphate pH 8.0, 300 mM NaCl, 500 mM imidazole, and fractions collected and analyzed on SDS-PAGE. Fractions containing the S protein were pooled, concentrated, and buffer exchanged to PBS using a 100-kDa MWCO concentrator. Proteins were flash frozen in liquid nitrogen and stored at -80 °C

### X-ray crystallography

Co-crystals of sRBD and E2 were generated by screening sRBD at 16 mg/mL in 20 mM Tris-Acetate pH 7.2, 200 mM NaCl and 4 mM E2 with the BCS screen (molecular dimensions). Drops were set up using the mosquito robotics system (SPT Labtech) at 0.2 μL protein solution 0.2 μL screen solution using the sitting drop vapor-diffusion method. Crystals were observed in a number of conditions in the BCS screen. The condition that yielded the crystals from which the final data set was collected grew in 3 days in 22 %v/v PEGSB (Precipitant) 0.1 M Na Phos Cit 5.5 pH (Buffer). The crystals were cryo-cooled in liquid nitrogen in the same solution for data collection. Co-crystals of sRBD and E2b were generated by screening sRBD at 16 mg/mL in 20 mM Tris-Acetate pH 7.2, 200 mM NaCl and 4 mM E2b with the BCS screen (molecular dimensions). Drops were set up using the mosquito robotics system (SPT Labtech) at 0.2 μL protein solution 0.2 μL screen solution using the sitting drop vapor-diffusion method. Crystals were observed in a number of conditions in the BCS screen. The condition that yielded the crystals from which the final data set was collected grew in 3 days in 22.5 %v/v PEGSM (Precipitant), 2 %v/v Glycerol (Precipitant), $0.01 M CoCl_2$ (Salt), $0.2 M MgCl_2$ (Salt), 0.1 M BIS-TRIS prop 8 pH (Buffer). The crystals were cryo-cooled in liquid nitrogen in the same solution for data collection. X-ray diffraction data were collected at Diamond synchrotron radiation sources (beamline i04), then processed using the pipedream package by Global Phasing Ltd; structures were solved using Phaser[55] from the CCP4 package[56]. Models were iteratively refined and rebuilt by using Refmac[57] and Coot programs[58]. Ligand coordinates and restraints were generated from their SMILES strings using the AceDRG[59] software from the CCP4 package[56].

## AlphaScreen epitope binning

Representative biotinylated *Bicycles* were incubated with the relevant poly(Histidine) tagged Spike protein domain (S2: ACROBiosystems, S2N-C52H5. S1: Novus Biologicals, NBP2-90985. Spike Trimer: pCOV53), along with a duplicate titration of multiple unmodified *Bicycles*. Finally, 20 μg/mL of AlphaScreen Streptavidin donor beads (Perkin Elmer) and 20 μg/mL Nickel-chelate AlphaLISA Acceptor beads (Perkin Elmer) were added and incubated. Plates were read on a Pherastar FS/FSX (BMG Labtech) using excitation wavelength 680 nm, emission wavelength 615 nm. Data were normalized to beads alone (low) and no competitor (high) reference averages. The normalized data was used to generate a four-parameter logistic curve fit in Dotmatics. Where no $IC_{50}$ was generated at the top concentration tested, this was used to define epitope boundaries.

## AlphaScreen Spike-hACE2 competition

Poly(Histidine) tagged Spike Trimer (pCOV54) at 0.20 nM was incubated with 0.20 nM of hACE2 Fc-fusion (ACROBiosystems - AC2-H5257) and a duplicate titration of monomeric or multimeric untagged *Bicycles*. Finally, 20 μg/mL Nickel-chelate Alpha donor beads (Perkin Elmer) alongside 20 μg/mL AlphaScreen Protein A acceptor beads (Perkin Elmer) were added and incubated. Plates were read on a Pherastar FS/FSX (BMG Labtech) using excitation wavelength 680 nm, emission wavelength 615 nm. Data was normalized to beads alone (low) and no competitor (high) reference averages. The normalized data were used to generate a four-parameter logistic curve fit in Dotmatics and GraphPad Prism v9.4.0 for additional information. Where no $IC_{50}$ was generated, this was reported as greater than the top concentration tested.

## Surface Plasmon Resonance

Protein constructs of Spike Trimer (pCOV54), S1 (Novus Biologicals - NBP2-90985), S1-RBD (ACROBiosystems – SPD-C82E9), S1-NTD (ACROBiosystems – S1D-C52H6), S2 (ACROBiosystems – S2N-C52E8) and ACE2 (ACROBiosystems – AC2-H52H8) were immobilized on a Series S CM5 Biacore T200 (Cytiva) sensorchip using standard primary amine coupling techniques to capture levels of approximately 1400 RU, 950 RU, 280 RU, 810 RU, 620 RU, and 450 RU, respectively. The carboxymethyl dextran surface was activated via a 1:1 v/v ratio of 0.4 M 1-ethyl-3-(3-dimethylaminopropyl) carbodiimide hydrochloride (EDC) / 0.1 M N-hydroxy succinimide (NHS). Spike Trimer, S1, S1-RBD, S1-NTD, S2 and ACE2 were diluted to 100 nM, 100 nM, 400 nM, 100 nM, 100 nM, and 100 nM respectively in 10 mM sodium acetate pH 5.0, pH 4.5, pH 4, pH 5.5, pH 5 and pH 5 as listed for coupling. Residual-activated groups were blocked using 1.0 M ethanolamine pH 8.5. Using an appropriate top concentration (10000 nM maximum), an 8-point titration of each *Bicycle* was screened against each protein construct using a flow rate of 50 μL/min. The running buffer was PBS-P+, 1.0% DMSO at pH 7.4. A multi-cycle kinetic format was used. Data were solvent corrected for DMSO bulk effects. All data were double-reference corrected against the reference flow cell and matched buffer blanks. Data processing and kinetic fitting were performed using Biacore T200 Evaluation Software (v3.2.1). Data were fitted using the 1:1 binding model or steady-state affinity model where appropriate. The geometric mean (geomean) $K_D$ (nM) for binders was calculated by transforming unique $K_D$ values by $log_{10}$, arithmetic mean averaging the transformed values, and subsequently antilog transforming the averaged value.

## Cells and viruses

Overall, 293 T cells for lentiviral production were obtained from ATCC (CRL-3216). 293 T ACE2/TMPRSS2 and Vero ACE2/TMPRSS2 were obtained from a previous study[42]. Briefly, virions were produced by transfection with 1 μg of the plasmid encoding SARS CoV-2 Spike protein (pCAGGS-SpikeΔc19), 1 μg pCRV GagPol and 1.5 μg GFP-encoding plasmid (CSGW)[60]. Viral supernatants were filtered through a 0.45 μm syringe filter at 48 h and 72 h post-transfection and either used as it is for WT PV and Omicron variant or pelleted for 2 h at 28,000x*g*. Pelleted virions were drained and then resuspended in DMEM (Gibco). SARS-CoV-2 isolate SARS-CoV-2/human/Liverpool/REMRQ0001/2020 was a kind gift from Lance Turtle (University of Liverpool) and David Matthews and Andrew Davidson (University of Bristol). Virus stock was generated in Vero ACE2/TMPRSS2 cells by infecting at low moi of 0.05 and incubating for three days. Supernatants were freeze/thawed three times, aliquoted and stored at −70 °C. Titres were assessed by plaque assay. All experiment with live SARS-CoV-2 were conducted in Containment Level 3 laboratory.

## Infections and RT-qPCR

Vero ACE2/TMPRSS2 cells were seeded into 96-well plates at density of $10^4$ per well day prior infection. *Bicycles* were serially diluted in FreeStyle 293 serum-free media, mixed with media containing the virus at moi=1 and incubated for 1 h. Solutions were then added to cells and incubated for 18 h. To quantify infections cells were freeze-thawed once and then lysed with 1 volume of lysis buffer (0.25% Triton X-100, 50 mM KCl, 100 mM Tris–HCl pH 7.4, glycerol 40% and RNAsecure (1/100)) for 5 minutes. Cell lysates were transferred to PCR plates and virus inactivated at 95 °C for 5 min. RT–qPCR was performed with Luna®Universal Probe One-Step kit (NEB) using final concentrations of 500 nM for each primer and 125 nM for the probes. Primer/probe for genomic viral RNA were CDC-N2 and N1 (from IDT) and 18 S control. For standards, we used SARS-CoV-2_N_Positive control RNA (from IDT) and 18 S DNA that was synthesized and kindly gifted by Jordan Clarks and James Stewart (University of Liverpool). RT–qPCRs were run on ABI StepOnePlus PCR System with following program: 55 °C for 10 min, 95 °C for 1 min and then 40 cycles of 95 °C denaturation for 10 s and 60 °C extension for 30 s. RNA copy numbers were obtained from standards and then genomic copies of N normalized to copies of 18 S. Finally, all data were normalized to 100% DMSO control and log transformed.

## Plaque assay for SARS-CoV-2

Cells were infected on 96-well plates as for RT-qPCR assay. After incubation, the plate was freeze-thawed three times. Vero ACE2/TMPRSS2 cells were seeded on 24-well plates day prior infection at $1.5 \times 10^5$ per well. Supernatants from 96-well plates were serially diluted and added to cells on 24-well plates. After 1 h incubation media containing 2% FBS and 0.05% agarose was added to cells and incubated for 3 days. Cells were fixed with 4% formaldehyde, stained with 0.1% toluidine blue and counted using phase contrast microscope. For lung homogenate from mouse challenge experiments, 10-fold serial dilutions of clarified lung homogenate were used to infect monolayers of Vero ACE2/TMPRSS2 cells, then the assay proceeded as above.

## Immunofluorescence

Vero ACE2/TMPRSS2 cells were seeded into 8-well plastic Ibidi dishes day prior infection at density of $4 \times 10^4$ per well. The following day *Bicycles* were diluted to 100μM in serum free media and virus stock added aiming at moi=1. LMB1 serum control (double vaccinated) was diluted 1 in 100. After 1 h incubation solutions were added to cells and incubated for a further 1 h. After that virus-containing media was removed and wells washed once with PBS. Cells were incubated for 18 h then fixed with 4% formaldehyde (Sigma) for 30 min and processed for immunofluorescence staining. Shortly thereafter, cells were permeabilised with 0.1% Triton X-100 in PBS for 5 min, blocked with 5% FBS and then labeled for nucleoprotein with SARS-CoV-2 Nucleocapsid antibody (anti-NP antibody) from Antibodies Online (https://www.antibodies-online.com/antibody/6953059/anti-SARS-CoV-2+Nucleocapsid+SARS-CoV-2+N+antibody/), Cat No: ABIN6953059, Dilution 1/500, validated by ELSA and western blot. Secondary antibody was then added; Alexa Fluor anti-rabbit 488 from Molecular

Probes (https://www.thermofisher.com/antibody/product/Goat-anti-Rabbit-IgG-Fc-Cross-Adsorbed-Secondary-Antibody-Polyclonal/A78953), Cat No: A78953, Dilution 1/2000, validated by Peptide array according to manufacturer. Nuclei were labeled with NucBlue from Invitrogen. Images were taken with Zeiss 780 microscope at magnification 20x.

### Spike-pseudotyped viral infection assays

Spike-pseudotyped viral infection assays were performed using HEK 293T-hACE2-TMPRSS2 cells, as described previously (33493182). In brief, cells were plated into 96-well plates at a density of $2 \times 10^3$ cells per well in Free style 293 T expression media and allowed to attach overnight. Overall, 18 μL pseudovirus-containing supernatant was mixed with 2 μL dilutions of *Bicycle* peptide and incubated for 40 min at RT. 10 μL of this mixture was added to cells. 72 h later, cell entry was detected through the expression of GFP by visualization on an Incucyte S3 live cell imaging system (Sartorius). The percentage of cell entry was quantified as GFP positive areas of cells over the total area covered by cells. Entry inhibition by the sera was calculated as percent virus infection relative to virus-only control.

### Cell–cell fusion assay

Cell–cell fusion assays were carried out as previously described (35104837). In brief, Vero ACE2/TMPRSS2 expressing either GFP1–10 or GFP11 were seeded at 80% confluence in a 1:1 ratio in a 96-well plate. Cells were transfected the next day with 0.1 μg per well of spike expression plasmids in pCDNA3 using Fugene 6 and following the manufacturer's instructions (Promega). *Bicycles* were added 5 h post-transfection. Cell–cell fusion was measured using Incucyte and determined as the proportion of green area to total phase area. Data were then analyzed using Incucyte software.

### Pharmacokinetic Studies

In vivo pharmacokinetic studies in male mouse (CD-1, 27 g – 38 g) and male Golden Syrian hamster (LVG, 95 g – 119 g) were conducted at Wuxi AppTec Co. Ltd. (Shanghai) using standard protocols based on composite blood sampling. All the procedures related to animal handling, care, and treatment in the studies were performed according to the guidelines approved by the Institutional Animal Care and Use Committee (IACUC) of WuXi AppTec (Shanghai, China), following the guidance of the Association for Assessment and Accreditation of Laboratory Animal Care (AAALAC). Briefly, animals were administered compounds either intravenously (15 min infusion) or subcutaneously at doses of 3.0 mg/kg and 10 mg/kg, respectively. Compounds were formulated as a clear solution containing 1 mg/mL (intravenous) or 2 mg/mL (subcutaneous) of each compound using 25 mM Histidine HCl buffer containing 10% (w/v) sucrose at pH 7.0. Plasma samples were obtained at regular time points up to 24 h postdose. A group size of 6 animals per study was used, 3 animals per time point with each animal sampled at alternate time points. Plasma samples were analyzed for BCY17021 and BCY18656 using qualified bioanalytical methods based on protein precipitation followed by liquid chromatographic triple quadrupole mass spectrometric (LC-MS/MS) analysis (LC-MS/MS-AY_Triple Quad 6500 + , SCIEX). Prepared plasma samples were spiked with internal standard and analyzed after methanol extraction. Standards and controls were prepared in plasma matrices in an identical manner. Pharmacokinetic parameters were calculated using Phoenix® WinNonlin® version 8.2 (Certara L.P. (Pharsight), St. Louis, MO).

### Pharmacokinetic-pharmacodynamic (PKPD) modeling

Target coverage plots were constructed using a simple sigmoid $E_{max}$ model based on unbound plasma concentrations of drug (corrected for protein binding), the $IC_{50}$ obtained for antiviral activity and an $E_{max}$ of 100% target coverage. The Hill Coefficient was constrained to a value of 1.0 as it was not rigorously investigated to prove co-operative functional activity. Plasma concentrations for each dose were estimated (assuming linear pharmacokinetics) using the measured plasma-concentration time profile following s.c dosing to each species, and then extrapolated to 24 h based on the terminal elimination-rate constant.

### Mouse SARS-CoV-2 challenge

Murine challenge studies were conducted in accordance with UK Home Office Scientific Procedures Act (ASPA, 1986) and project licence PP4715625 has been approved by the University of Liverpool Animal Welfare and Ethical Review Board (AWERB). Male K18 hACE2 mice were purchased from Charles River (18-20 g). Mice were housed in groups of 4 to 5. Photoperiod = 12 on:12 off dark/light cycle. Ambient animal room temperature is 21 °C, controlled within ± 2 °C and room humidity 50% ± 5%. On day -1 relative to infection, mice were treated by subcutaneous injections three times per 24 h period, every 8 h (t.i.d.), except remdesivir which was subcutaneously administered twice per 24 h period, every 12 h *bis in die* (b.i.d.). On Day 0, mice were infected intranasally with $10^4$ PFU/mouse SARS-CoV-2/human/Liverpool/REMRQ0001/2020 using 50 μL inoculum, 25 μL per nostril. Viral inoculum was administered post the concurrent drug or vehicle (25 mM Histidine HCl buffer, 10% (w/v) sucrose at pH 7.0) dose. Mice continued to be treated t.i.d. with either vehicle, *Bicycle* or a positive control drug, remdesivir until Day 4 at which point the animals were culled. An uninfected control group of mice was also added, receiving vehicle only (t.i.d.). A treatment group size of 4 animals was used. E2 trimer *Bicycles* were administered at 300 mg/kg t.i.d. (900 mg/kg/day), E2E4 *Bicycles* were administered at 100 mg/kg t.i.d. (900 mg/kg/day), remdesivir was administered at 25 mg/kg BID (50 mg/kg/day). Four days postinfection, animals were culled using an overdose of pentobarbitone and lungs and nasal turbinates were removed for qPCR and plaque assay. Lung tissue was homogenized using a Bead Ruptor 24 (Omni International) then clarified by centrifugation at 12,000 x g for 5 min. Viral RNA derived from nasal turbinates were quantified using a protocol adapted from the CDC 2019-Novel Coronavirus (2019-nCoV). Real-Time PCR was used to measure sub-genomic viral RNA to the E gene (sgE). The GoTaq® Probe 1-Step RT-qPCR System was utilized for both protocols and data were normalized to 18 S data for subsequent quantification.

### qPCR for cytokines and chemokines

For qPCR using probes against cytokines and chemokines, total cellular RNA from lung or nasal turbinate was isolated using the RNeasy minikit (Qiagen), including DNase digestion (RNase-free DNase kit; Qiagen). One microgram of total cellular RNA was reverse transcribed into cDNA by using Superscript II (Invitrogen) and oligo dT and used for quantitative real-time PCR. Gene expression was monitored by TaqMan Gene Expression Assays (Applied Biosystems) on a StepOnePlus Real Time PCR System (Life Technologies). Taqman gene expression assay mixes were as follows: mouse β-actin (4352933E), IFN-β1 (Mm00439552_s1), CxCL10 (Mm00445235_m1), TNFα (Mm00443260_g1), CCL2 (Mm00441242_m1), CCL3 (Mm00441258_m1), CCL4 (Mm00443111_m1), CCL5 (Mm01302427_m1), CxCL9 (Mm00434946_m1), IL-6 (Mm00446190_m1), IFNg (Mm01168134_m1), Mx1 (Mm00487796_m1). Relative gene expression compared to actin reference gene was determined using the change-in-threshold ($2^{-\Delta\Delta CT}$) method.

### Syrian golden hamster SARS-CoV-2 challenge

Hamster challenge studies were carried out in the central animal facilities of Viroclinics Biosciences B.V. in Schaijk, The Netherlands, under conditions that meet the standard of Dutch law for animal experimentation and are in agreement with the "Guide for the care and use of laboratory animals", Institute for Laboratory Animal Research (ILAR) recommendations, AAALAC standards. Ethics approval was made by the local animal welfare body for the present study and registered under

number: 27700202114492-WP16. On day -1, at -4 h relative to infection, male Syrian golden hamsters (Janvier, 94 g – 112 g) were treated by subcutaneous injections three times per 24 h period, every 8 h (t.i.d.). On Day 0, hamsters were infected intranasally with $10^2$ $TCID_{50}$ BetaCoV/Munich/BavPat1/2020 using 100 μL inoculum, 50 μL per nostril. Viral inoculum was administered 4 hours post the initial treatment administration. Hamsters continued to be treated t.i.d. with either vehicle (25 mM Histidine HCl buffer, 10% (w/v) sucrose at pH 7.0) or *Bicycle* until Day 4. A treatment group size of 6 animals was used. Homotrimer *Bicycle* was administered at 100 mg/kg t.i.d. (300 mg/kg/day), Biparatopic *Bicycle* was administered at 1, 3 and 10 mg/kg t.i.d. (3, 9 or 30 mg/kg/day). On Day 4 postinfection, animals were culled and lungs taken for analysis. Serial dilutions of lung homogenate were incubated on Vero E6 cell monolayers for 1 hour at 37 °C. Vero E6 monolayers were washed and incubated for 4-6 days at 37 °C after which plates were scored using the vitality marker WST8 (colorimetric readout). WST-8 stock solution was prepared and added to the plates. Per well, 20 μL of solution (containing 4 μL of the ready-to-use WST-8 solution from the kit and 16 μL inoculation medium, 1:5 dilution) was added and incubated 3-5 hours at RT. Subsequently, plates were measured for optical density at 450 nm (OD450) using a microplate reader and visual results of the positive controls (cytopathic effect (CPE)) were used to set the limits of the WST-8 staining (OD value associated with CPE). Viral titers ($TCID_{50}$) were calculated using the method of Spearman-Karber. Data was collected in quadruplicate.

### Reporting summary

Further information on research design is available in the Nature Portfolio Reporting Summary linked to this article.

## Data availability

All data are available in the manuscript, in accompanying figures and tables and in the accompanying Source Data file, with the exception of two x-ray structures that have been deposited in the PDB database (https://www.rcsb.org/) with codes 7Z8O and 8AAA and also three previously deposited structures from the PDB database with codes 6M0J, 7A98 and 6ZP0.

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

## Acknowledgements

This work was supported by the MRC (UK; U105181010), a Wellcome Trust Investigator Award (223054/Z/21/Z), a Wellcome Trust Collaborator Award (214344/A/18/Z) and Innovate-UK (UKRI Ideas to Address COVID-19 – Innovate UK Article 25 funding strand). AO and JPS acknowledge funding from EPSRC, Unitaid, Wellcome Trust and MRC for funding which supported development of preclinical models for SARS-CoV-2 infection. We would like to thank Viroclinics Xplore for in vivo studies, Evotec for contract research services, Rachel Dods and Gustavo Arruda Bezerra for structural analysis, Radu Aricescu for design of lentiviral S protein expression systems, Andrew Carter for cloning of S-pseudotyped vectors, Dean Clift for Incucyte analysis, Tyler Rhinesmith and Jakub Laptuk for cell culture and assay support and the Bicycle team and James lab for general scientific support.

## Author contributions

M.J.S. and L.C.J. provided project and scientific direction. L.C.J. wrote the original manuscript and L.C.J., M.J.S., K.U.G., M.A.J.H., P.B., L.C., J.A.G.B. & A.O. revised and edited. Authors contributed to original research and data analysis as follows: K.U.G. and L.C.: bacteriophage screening, M.V.: CL1 and CL2 infections and assays, M.A.J.H.: pharmacology data, A.A.: CL3 infections and assays, P.J.: pharmacokinetic modeling, P.B.: chemistry strategy, G.P.: Syncytia formation data, D.M.: S-pseudovirus assay creation, B.M.: in vivo study design, K.V.R. and S.S.: *Bicycle* design and synthesis, P.B., A.L. and M.H.: Structure determination, K.C. & V.T.C.: Spike trimer engineering and production, J.P.S., A.O., J.S., M.N., H.B., J.H., E.K., L.T., E.G.B., P.S., A.K. & X.H.: mouse challenge.

## Competing interests

The authors declare no competing nonfinancial interests. The authors declare no competing financial interests except: K.U.G., M.A.J.H., P.J., P.B., L.C., B.M., K.V.R., S.S. and M.J.S. who are shareholders in Bicycle Therapeutics Plc, the parent company of BicycleTx Ltd.
