## [Peer Review File · Nature Communications]

REVIEWER COMMENTS

Reviewer #1 (Remarks to the Author):

This study describes the development of bicyclic peptides binding to SARS-CoV-2 spike protein of which several prevent viral infection. The bicyclic peptides were identified by screening large libraries using phage display. Binders were found to up to 10 different epitopes of the spike protein. One family was directly inhibiting the binding of RBD:ACE2 and thus the binding of the spike protein to ACE2. The structure of two peptides of the latter group, bound to ACE2, were solved by X-ray crystallography. Multimerizing the bicyclic peptides improved their binding affinity from rather weak binders to potent nanomolar inhibitors. Biparatopic bicyclic peptide conjugates were also developed and found to be even more potent ($IC_{50} = 1nM$). Finally, trimeric and biparatopic bicyclic peptides were tested in mouse and hamster models and were shown to efficiently prevent SARS-CoV-2 infection.

This is a large, comprehensive study, starting with the development of a wide range of specific peptide-based ligands, continuing with an in depth characterization (incl. X-ray structure), then the optimization via multimerization, and culminating in the successful testing in vivo. The study appears of good quality (with some data missing in the text or in figures, as described below). The bicyclic peptides (and the multimers) are rapidly cleared in vivo and thus needed to be dosed by injection and multiple times per day in the study, which is a disadvantage over small molecules (that are oral) and monoclonals (that need to be doses once per week or even less), which should be highlighted more. I still think that the modality of bicyclic peptides is of interest for preventing SARS-CoV-2 infection and has some potential (e.g. the bicyclic peptides may be conjugated to antibodies to address the fast clearance) and this study thus highly relevant. I can recommend publication if the following points can be addressed.

Criticism:

1. Title and manuscript: The term "Bicycles" is not ideal as it sounds almost like a trade mark name (which is not ideal as many of the authors are from Bicycle Therapeutics). I recommend to change to "bicyclic peptides" which is used in scientific literature to describe this modality.
2. Introduction: The authors write that bicyclic peptides combine advantages of small molecule drugs and monoclonal antibodies, which is not a complete description. They combine also several disadvantages, which should also be discussed, such as the short half-life and the need for injection.
3. The peptides sequences and consensus groups should be shown in a main figures (aligned).

4. The Kds of all monomeric peptides should be indicated in a main figure.
5. Abbreviations: GTA, TCA, TET should be described.
6. IC50 values of all bicyclic peptides, multimers and biparatopics should be indicated in the text when described.
7. NGS data should be provided.

Reviewer #2 (Remarks to the Author):

In this article, authors describe the generation of a novel antiviral approach targeted for SARS-CoV-2, and efficacy testing of said approach in several in vitro and in vivo models. There is a continuous need for novel antiviral approaches against RNA viruses, including but not limited to SARS-CoV-2, and efforts to establish new treatment modalities should be commended. Authors employ a range of structural, biochemical, and virological strategies to draw conclusions regarding efficacy of the bicyclic peptide approach evaluated here, and do include limited efficacy data against a range of SARS-CoV-2 variants of concern. The manuscript is well-written and generally logically organized. However, there are methodological details missing from several experimental designs, with some conclusions drawn from experiments that are not fully supported by the data as currently presented. Furthermore, the discussion section is lacking sufficient context with previously published literature.

Comments:

-It is unclear why authors chose not to include section headers throughout the results; formatting to include this traditional structure would improve readability of the text overall.

-Authors state that "...bicycles potentially inhibited infection (Figure 3C)" employing an immunohistochemistry approach, but the supporting microscopy images do not sufficiently support this. All fields show green NP-stained cells; authors should quantify the percentage of infected cells following treatment under each condition (e.g. by counterstaining with DAPI, or by reporting a total fluorescence intensity between groups) to more quantifiably support this statement.

-Similarly, data in Figure 3 E-F does not sufficiently support conclusions drawn in its current form. In Figure 3E, data with bicycle efficiency is shown as changes to "relative cell-cell fusion" but the authors do not disclose what fusion levels are relative to. In Figure 3F, authors present microscopy images showing no syncytia formation with bicycle administration relative to no treatment, but do not state the concentration of bicycle employed in this assay, and do not include sufficient controls (e.g. showing concentrations of treatment that do/do not induce syncytia formation, to show that this phenotype is dependent on the concentration of treatment employed (like what is shown in Fig 3E) and not a byproduct of the treatment condition itself).

-The methods section is lacking information in some sections, particularly with regard to the in vivo experiments. Please specify the limit of detection for infectious virus plaque assay readouts (for both infectious virus assays employed in this study, this information is missing in both spots). Figure 5C shows 0 PFU/ml in selected groups but it is very unlikely the LOD was actually 0 (while there is a dashed line in some panels of this image, the line is not specifically identified as the LOD. For cell-cell fusion assays, please specify the concentration of bicycles added post-transfection. For both in vivo challenges, please specify the gender of all animals, volume of inoculum administered to animals, and specify the timing of dosing of drug (text states 3 times per 24 hr period prior to infection, but does not state the interval, is this every 8 hours? And please specify if the 3x/day regime continued throughout the experiment and at what intervals). If treatment was administered concurrent with viral inoculation, please state the order in which these events was conducted. For pharmacokinetic studies, please specify the gender of both animals employed, and provide a reference for "standard protocols" that were employed for this testing. For efficacy, host response, and pharmacokinetic studies, please specify group sizes explicitly in the results and/or figure legends.

-For statistics comparing groups where values appear below the limit of detection, authors need to state what the null value is assigned to them for statistical purposes (Figures 4B and 4C would suggest these are 0, but they should be set at the limit of detection of the assay). Either way this should be more clearly presented to the reader.

-What were viral loads in the nasal turbinates of infected and treated animals? It is currently unclear why authors employed PCR-based detection methods for the nasal tissue (Fig 5A-B) and infectious titer readouts for the lung tissue (Fig 5C) instead of using both approaches uniformly for all tissues collected.

-For host response studies (Fig5D-F), please state if “uninfected” animals were mock-inoculated with vehicle-only or not prior to tissue collection.

-The discussion section is rather abbreviated and does not touch on several areas warranting inclusion. In fact, there are no references to previously published literature at all in the discussion section; this is a major limitation to the paper as currently written. Examples where expansion of text would be warranted include: anticipated pharmacokinetic profile of this treatment approach in humans (current text acknowledges the limitation in directly comparing results from mammalian models to humans but provides no subsequent context in how to interpret it), how successful these treatment approaches would be beyond day 4 post-inoculation in preventing severe disease/lethal outcomes, applicability of this design approach to other viruses of public health importance (the potential for this is mentioned briefly but additional context and references would be of benefit), etc.

Reviewer #1

Q1. Title and manuscript: *The term "Bicycles" is not ideal as it sounds almost like a trade mark name (which is not ideal as many of the authors are from Bicycle Therapeutics). I recommend to change to "bicyclic peptides" which is used in scientific literature to describe this modality.*

A1. We have used the term 'Bicycles' to be consistent with the previously published literature on these molecules, eg Recent papers from Mudd et al, 2020 & 2022, Hurov et al 2021, Upadyaya et al 2022 and Stanway et al 2022 all reference Bicycles within the title and/or text because they are chemically distinct from other types of bicyclic peptides. To address this point we have added a molecular definition of 'Bicycles' and explained why they are different from other bicyclic peptides: "*Bicycles* are structurally and chemically distinct from other types of bi-cyclic peptides and therefore we use this term specifically to refer to cyclized linear peptides with a symmetric, central small-molecule scaffold that forms covalent thioether bonds with three cysteines encoded in the linear peptide sequence".

Q2. Introduction: *The authors write that bicyclic peptides combine advantages of small molecule drugs and monoclonal antibodies, which is not a complete description. They combine also several disadvantages, which should also be discussed, such as the short half-life and the need for injection.*

A2. This important point has been addressed within the text, the reviewer rightly points out that a short-systemic half-life could be perceived as a limitation of these molecules, however there are precedented Bicycle solutions to extend half-life that could be deployed in the event that this became a limiting factor. In our experience, when Bicycles are formed as multimers this has a significant effect on extending human and non-human primate PK, often to multiple hours. We have therefore added the following to the introduction: "Potential disadvantages when compared to small molecules, such as a short systemic half-life can be rectified through simple chemical modifications or multimerization^{33,34,37,38} and low oral bioavailability through delivery via alternative routes, such as subcutaneous, inhalation or intra-nasal dosing." Clearly, we have yet to progress the compounds described in this publication into higher species and so cannot make this claim here but we would predict a similar extension in half-life. Bicycles also have low oral bioavailability. In other studies, we have shown that Bicycles can be administered via topical (inhaled and intranasal [WO 2021/205161A1]) as well as subcutaneous routes. As has been shown with antibodies and biologic drugs, this inability to be delivered via oral delivery has not impeded the clinical progression of these classes of drugs for either SARS-CoV-2 or other important diseases, nor their commercial success where 9 out of the top 10 selling therapeutics in 2022 were delivered via the parenteral route of administration.

Q3. The peptides sequences and consensus groups should be shown in a main figures (aligned).

A3. We have created a new table (Table 1) listing both the exemplar sequence for each epitope group used in the study and the consensus sequence. Note that we used Sanger sequencing and not NGS to screen our hits. During phage screening, thousands of clones were sanger sequenced, resulting in identification of multiple replicates, incorrect clones, and non-specific binders. Binding phage clones were identified by AlphaScreen assay, chemically synthesized as peptides, and re-screened against protein using SPR to check if they were real binders. Here we only show the exemplar peptide sequence for each of the epitopes, that has been determined to be a real binder. In addition, where known, we have indicated what the consensus sequence of key binding residues is for each of these peptides

Q4. The Kds of all monomeric peptides should be indicated in a main figure.

A4. We have reformatted Figure 1C to include the “Geomean K_D (nM)” value where binding was observed to a given isolate Spike Protein sub-domain. The full numerical summary remains available in Supplementary Table 1.

Q5. *Abbreviations: GTA, TCA, TET should be described.*

A5. The abbreviations for GTA - Glutaric Acid, TCA – Tri(carboxyethoxyethyl)amine and TET - 1,3-bis(carboxyethoxy)-2,2-bis(carboxyethoxy)propane, have now been included in the main text. We also identified a similar scenario, replacing the BAPG hinge abbreviation with the unabbreviated N-(acid-PEG3)-N-bis(PEG3-amido)-PEG36-triazolyl linker.

Q6. *IC50 values of all bicyclic peptides, multimers and biparatopics should be indicated in the text when described.*

A6. We have added IC50 values into the main results and figure legends as requested. IC90 values are also given for every Bicycle in a separate table, Supplementary Table 6.

Q7. *NGS data should be provided.*

A7. NGS was not performed. All sequencing was carried out by sanger sequencing techniques and this has now been clarified in the methods section: “After round 4, phage clones were isolated, sanger sequenced, and tested for binding to proteins using *AlphaScreen* assay”. Table 1 (now added to main section) describes the sequences of key peptides synthesized and tested in assays.

Reviewer #2

Q1. *It is unclear why authors chose not to include section headers throughout the results; formatting to include this traditional structure would improve readability of the text overall.*

A1. We thank the reviewer for this suggestion and have added section headers.

Q2. *Authors state that “...bicycles potentially inhibited infection (Figure 3C)” employing an immunohistochemistry approach, but the supporting microscopy images do not sufficiently support this. All fields show green NP-stained cells; authors should quantify the percentage of infected cells following treatment under each condition (e.g. by counterstaining with DAPI, or by reporting a total fluorescence intensity between groups) to more quantifiably support this statement.*

A2. We have counterstained each condition with DAPI (shown in grey in the IF images) and used Fiji image processing to automatically count cells and quantify the number that are NP positive. The resulting data is now included as Supplementary Figure 1.

Q3. *Similarly, data in Figure 3 E-F does not sufficiently support conclusions drawn in its current form. In Figure 3E, data with bicycle efficiency is shown as changes to “relative cell-cell fusion” but the authors do not disclose what fusion levels are relative to. In Figure 3F, authors present microscopy images showing no syncytia formation with bicycle administration relative to no treatment, but do not state the concentration of bicycle employed in this assay, and do not include sufficient controls (e.g. showing concentrations of treatment that do/do not induce syncytia formation, to show that this phenotype is dependent on the concentration of treatment employed (like what is shown in Fig 3E) and not a byproduct of the treatment condition itself).*

A3. We have now provided the primary immunofluorescence data from which Figure 3E&F is derived. The immunofluorescence data for every data point (i.e. every Bicycle concentration) is now shown in Supplementary Figure 1B. The concentration of Bicycles used in the exemplar IF images in Figure 3F is also now stated in the figure legend. We have also updated the legend to explain that relative cell-cell fusion refers to the level of fusion relative to that observed in the absence of Bicycle (ie DMSO only condition). Under DMSO only conditions, approximately 30% cells undergo fusion, which we have set as the maximum possible degree of syncytia formation (ie 1).

Q4a. *The methods section is lacking information in some sections, particularly with regard to the in vivo experiments. Please specify the limit of detection for infectious virus plaque assay readouts (for both infectious virus assays employed in this study, this information is missing in both spots). Figure 5C shows 0 PFU/ml in selected groups but it is very unlikely the LOD was actually 0 (while there is a dashed line in some panels of this image, the line is not specifically identified as the LOD. For cell-cell fusion assays, please specify the concentration of bicycles added post-transfection.*

A4a. We thank the reviewer for highlighting these issues. We have now corrected the figure legends to specify the LOD for each assay. For plaque assays, the limit of detection was $\log_{10}(0.82)$ pfu/ml. For lung homogenate cytopathic effect assays, the limit of detection ranged between 1.2 to 1.4 $\log_{10}(\text{TCID}_{50}/\text{g})$. We have also stated the concentration of Bicycles added post-transfection in the IF images shown in Figure 3F (32 nM) and have provided IF images for the full range of Bicycle concentrations shown in Figure 3E as new Supplementary Figure 1. Finally, the methods section for the in vivo experiments has been expanded as detailed in the response below.

Q4b. *For both in vivo challenges, please specify the gender of all animals, volume of inoculum administered to animals, and specify the timing of dosing of drug (text states 3 times per 24 hr period prior to infection, but does not state the interval, is this every 8 hours? And please specify if the 3x/day regime continued throughout the experiment and at what intervals). If treatment was administered concurrent with viral inoculation, please state the order in which these events was conducted. For pharmacokinetic studies, please specify the gender of both animals employed, and provide a reference for "standard protocols" that were employed for this testing. For efficacy, host response, and pharmacokinetic studies, please specify group sizes explicitly in the results and/or figure legends.*

A4b. We thank the reviewer for highlighting these areas lacking clarity, we have included additional methodology to address as follows:

- We have provided additional methodology to describe the standard protocols mentioned for pharmacokinetic studies, as opposed to a reference.
- All animals used for pharmacokinetic and efficacy studies were male and we have included animal mass ranges additionally.
- Bicycle and vehicle treatment conditions were administered every 8 h (TID) throughout each entire efficacy study.
- Viral inoculum was administered post the concurrent treatment condition.
- The volume of viral inoculum and the per nostril split has now been stated.
- Sample sizes have now been specified in the methodology and figure legends.

Q5. *For statistics comparing groups where values appear below the limit of detection, authors need to state what the null value is assigned to them for statistical purposes (Figures 4B and 4C would suggest these are 0, but they should be set at the limit of detection of the assay). Either way this should be more clearly presented to the reader.*

A5. We apologise for this error and as per the reviewers suggestion have set null values to the LOD and provided these in the figure legend.

Q6. What were viral loads in the nasal turbinates of infected and treated animals? It is currently unclear why authors employed PCR-based detection methods for the nasal tissue (Fig 5A-B) and infectious titer readouts for the lung tissue (Fig 5C) instead of using both approaches uniformly for all tissues collected.

A6. We were not able to measure viraemia in nasal turbinates. The different assays are unfortunately a consequence of the pandemic and the multi-site nature of the study. All materials had to be handled at CL3 and SOPs and RAs were being developed in real-time. For instance, qPCR measurements were possible in Liverpool, where the mouse challenges took place, but not plaque assays. We were able to arrange shipment of lung homogenate for plaque studies in Cambridge but there was not available turbinate material for similar processing. Rather than omit data we have opted to include everything we collected but we apologise that this means there is not the uniformity we would ideally like.

Q7. *For host response studies (Fig5D-F), please state if “uninfected” animals were mock-inoculated with vehicle-only or not prior to tissue collection.*

A7. Uninfected animals were given a vehicle-only control. We have added this to the revised manuscript's methodology and figure legend.

Q8. *The discussion section is rather abbreviated and does not touch on several areas warranting inclusion. In fact, there are no references to previously published literature at all in the discussion section; this is a major limitation to the paper as currently written. Examples where expansion of text would be warranted include: anticipated pharmacokinetic profile of this treatment approach in humans (current text acknowledges the limitation in directly comparing results from mammalian models to humans but provides no subsequent context in how to interpret it), how successful these treatment approaches would be beyond day 4 post-inoculation in preventing severe disease/lethal outcomes, applicability of this design approach to other viruses of public health importance (the potential for this is mentioned briefly but additional context and references would be of benefit), etc.*

A8. We appreciate the opportunity to expand the discussion of how Bicycles could be further developed for human use and to treat other viruses. We have therefore indicated potential future directions while trying not to speculate too far beyond the current data. In providing a useful context for pharmacokinetic profiles, we have drawn a comparison to the closest equivalent drug modality, Designed Ankyrin Repeat Proteins (DARPs) and added the following paragraphs to the discussion:

“The design principles of our multimeric Bicycles are broadly similar to that of the clinical biologic, Ensovibep⁴⁴⁻⁴⁶. Ensovibep encompasses three distinct SARS-CoV-2 RBD-binding Designed Ankyrin Repeat Proteins (DARPs) and two distinct human serum albumin (HSA)-binding DARPs. Ensovibep demonstrated anti-SARS-CoV-2 efficacy in both prophylactic and therapeutic preclinical *in vivo* models^{44,46}. This compound has recently completed early clinical evaluation and has shown therapeutic benefit in preventing infection, warranting further Phase II evaluation⁴⁷. This data is encouraging for the use of non-small molecule, non-antibody based therapies and suggests that pharmacokinetics need not be an obstacle to use in humans. However, the *in vivo* pharmacokinetic data presented in the current manuscript for *Bicycles* was generated to support the *in vivo* study design and interpretation of the results rather than predict human pharmacokinetics and simulation of the human plasma concentration-time profile. Future work will be required in other species, including non-human primates, which have previously been shown to provide reliable estimates of human pharmacokinetic parameters for peptide-based therapeutics⁴⁸.”

“Further experiments will also be required to explore treatment efficacy beyond 4 days. However, we are encouraged by the fact that by 4 days viraemia was undetectable in some conditions suggesting that sterile immunity had been achieved. The correlation between viral clearance and suppression of an inflammatory response we observed is also encouraging for positive patient outcomes. Moreover, this correlation, combined with the ready availability of diagnostic tests for SARS-CoV-2, suggests that viraemia represents a viable clinical and therapeutic endpoint. All viruses must interact with surface receptors to infect cells and the ability of neutralizing antibodies to prevent infection by inhibiting this interaction demonstrates the applicability of such an antiviral mechanism to diverse viruses of public health importance. Given that this is also one mechanism by which the Bicycles described here prevent SARS-CoV-2 infection, it seems reasonable to suppose that Bicycles developed against the cell-surface receptor binding proteins of other viruses would be useful to treat other viral infections. DARPins have been successfully developed to inhibit HIV-1⁴⁹ infection as well as SARS-CoV-2⁴⁵ and have been proposed as broad antivirals on the basis of inhibiting receptor binding⁴⁴. One potential benefit of *Bicycles*, as illustrated here, is the speed with which specific antiviral molecules can be generated. This property makes Bicycles ideal as an antiviral modality in the event of future pandemics caused by new or newly zoonosed viruses.

”

REVIEWER COMMENTS

Reviewer #1 (Remarks to the Author):

First of all, I like to reinforce that this work is substantial and could be of interest to a large audience, and I recommend considering it for publication in a high-impact journal like Nature Communications. The authors have addressed some points that I had criticized but some were not sufficiently addressed, or not at all, as described in the following (using the same numbers for the questions/answers used by the authors):

Q1/A1: Not addressed satisfactorily. I still think that it is not ideal to use the term "Bicycle" as it is mostly used by Bicycle Therapeutics. Also, the entire title sounds a bit like a title of an advertisement brochure of the company. In addition, the title states "therapeutics" which does not apply for the described molecules, that are not approved drugs. Given that most authors are from Bicycle Therapeutics, and the potential conflict of interest, I strongly recommend to change the title. E.g. what about "Multivalent bicyclic peptides binding SARS-CoV-2 protein show antiviral activity".

Q2/A2: Addressed partially. The introduction is improved by mentioning also limitations of bicyclic peptides. However, overall, the view appears still biased to me. E.g. the half live of bicyclic peptides is much shorter than that of mAbs and even with the best available technology cannot be improved to something comparable to the half-life of mAbs. I recommend to choose a more objective description, for the same reasons as described above (potential conflict of interest).

Q3/A3: Addressed partially. The added table 1 is ok. The raw data, being all the sequences found, aligned as consensus sequences, are still missing. This raw data is needed as this is core data of this study. It is required for the quality of the paper, if not shown in main figure, then as a SI figures. The fact that the phage were sequenced by Sanger sequencing makes it easier to display them.

Q4/A4: Addressed partially. I need to say that I do not know what "geomean Kd" is. The authors should describe what it is so that I can tell if I consider it as suited. Most preferred would be to simply indicate a proper Kd.

Q5/A5: well addressed.

Q6/A6: well addressed.

Q7/A7: Partially addressed: All sequences found by Sander sequencing should be provided. Those peptides that share a similarity should be aligned as described in Q3.

Reviewer #2 (Remarks to the Author):

The authors have addressed the majority of comments raised during peer review, though the files I had access to did not reflect the following changes which, per the responses to reviewers, were supposed to have been made:

-addition of results section sub-headings

-modifying the negative values in Figure 5B-C at the LOD level and not 0, with stats re-performed accordingly

Additionally, it would be of benefit to state the limitation of not being able to report infectious virus loads of nasal turbinates of infected vs treated animals (Q6, reviewer 2); it is likely other readers would have a similar question and the response to this query by the authors did not result in textual modifications within the manuscript to address it.

REVIEWER COMMENTS

Reviewer #1 (Remarks to the Author):

First of all, I like to reinforce that this work is substantial and could be of interest to a large audience, and I recommend considering it for publication in a high-impact journal like Nature Communications. The authors have addressed some points that I had criticized but some were not sufficiently addressed, or not at all, as described in the following (using the same numbers for the questions/answers used by the authors):

Q1/A1: Not addressed satisfactorily. I still think that it is not ideal to use the term "Bicycle" as it is mostly used by Bicycle Therapeutics. Also, the entire title sounds a bit like a title of an advertisement brochure of the company. In addition, the title states "therapeutics" which does not apply for the described molecules, that are not approved drugs. Given that most authors are from Bicycle Therapeutics, and the potential conflict of interest, I strongly recommend to change the title. E.g. what about "Multivalent bicyclic peptides binding SARS-CoV-2 protein show antiviral activity".

A1: We understand the sensitivity around the title and are happy to modify it. However, we feel it is important the title captures that bicyclic peptides are a different kind of antiviral format to antibodies and small molecules, rather than suggesting we've found a specific peptide that happens to have SARS-CoV-2 activity. We hope people will be interested in the idea that bicyclic peptides provide an alternative choice when developing antivirals in the future. Therefore we have amended the title to: "Multivalent bicyclic peptides are an effective antiviral modality that can potently inhibit SARS-CoV-2".

Q2/A2: Addressed partially. The introduction is improved by mentioning also limitations of bicyclic peptides. However, overall, the view appears still biased to me. E.g. The half live of bicyclic peptides is much shorter than that of mAbs and even with the best available technology cannot be improved to something comparable to the half-life of mAbs. I recommend to choose a more objective description, for the same reasons as described above (potential conflict of interest).

A2: We have added additional text on Bicycle limitations: "Antibodies also have extremely long half-lives and although it may not be possible to endow *Bicycles* with comparable longevity, serum persistence can be improved through simple chemical modifications or multimerization^{33,34,37,38}." And: "As with any modality, *Bicycles* will have limitations. Some like low oral bioavailability may be addressable through delivery via alternative routes, such as subcutaneous, inhalation or intra-nasal dosing."

However, we would like to note that the purpose of this section of the introduction is to explain why the study was undertaken. It therefore necessarily focuses on potential advantages that *Bicycles* could have as an antiviral modality.

Q3/A3: Addressed partially. The added table 1 is ok. The raw data, being all the sequences found, aligned as consensus sequences, are still missing. This raw data is needed as this is core data of this study. It is required for the quality of the paper, if not shown in main figure, then as a SI figures. The fact that the phage were sequenced by Sanger sequencing makes it easier to display them.

A3: An additional table has been added to SI that contains the additional aligned sequences, as well as indicating the lead sequence. Note, for epitope 6 and 7, these were identified as "singleton clones" without any similar sequences.

Q4/A4: Addressed partially. I need to say that I do not know what "geomean Kd" is. The authors should

describe what it is so that I can tell if I consider it as suited. Most preferred would be to simply indicate a proper K_d .

A4: Geometric Mean (Geomean) K_D (M) is equivalent to Arithmetic Mean pK_D . It is necessary to use a geometric mean as opposed to an arithmetic mean given the lognormal, (as opposed to Gaussian normal), distribution of affinity constants. This gives a better approximation of the central tendency of a lognormal dataset. A geometric mean simply transforms each value to be averaged by \log_{10} , the transformed values are then averaged and subsequently antilog transformed to produce the geometric mean average. A short summary of the geometric mean transformation is now indicated in Figure 1C's legend as follows: "Geomean averages were calculated by transforming unique K_D values by \log_{10} , arithmetic mean averaging the transformed values, and subsequently anti-log transforming the averaged value."

5. Q5/A5: *well addressed.*

6. Q6/A6: *well addressed.*

Q7/A7: *Partially addressed: All sequences found by Sander sequencing should be provided. Those peptides that share a similarity should be aligned as described in Q3.*

A7: As mentioned in response to Q3, all sequences have now been provided as a supplementary file.

Reviewer #2 (Remarks to the Author):

The authors have addressed the majority of comments raised during peer review, though the files I had access to did not reflect the following changes which, per the responses to reviewers, were supposed to have been made:

Q1: *addition of results section sub-headings.*

A1: We have added subheadings to match the data as presented in the Figures.

Q2: *modifying the negative values in Figure 5B-C at the LOD level and not 0, with stats re-performed accordingly.*

A2: We think perhaps the updated version of Figure 5 was not provided to the reviewer. We have now re-uploaded the correct version where the LOD of Figure 5B is 5 copies/ μg RNA (which on the \log_{10} axis is 0.7) and Figure 5C is 6.6 pfu/ml (which on the \log_{10} axis is 0.82). The stats have been calculated automatically in Prism using a one-way ANOVA.

Q3: *Additionally, it would be of benefit to state the limitation of not being able to report infectious virus loads of nasal turbinates of infected vs treated animals (Q6, reviewer 2); it is likely other readers would have a similar question and the response to this query by the authors did not result in textual modifications within the manuscript to address it.*

A3: We have added the following to the manuscript to explain the absence of plaque data for the nasal turbinates: "We were unfortunately unable to perform plaque assays on nasal turbinates due to challenges in processing and shipping samples between labs during the pandemic. However, we were able to determine whether *Bicycle* treatment reduced the levels of infectious virus in the lung by performing a series of plaque assays on lung homogenate."

REVIEWERS' COMMENTS

Reviewer #1 (Remarks to the Author):

The authors have addressed all concerns and have made changes. Specifically, they have changed the naming of the molecules, added all peptide sequence data, and they added clarifications. I can recommend publication.